# Faster Neural Network Training with Approximate Tensor Operations

**Menachem Adelman**
Intel & Technion
adelman.menachem@gmail.com

**Kfir Y. Levy** *
Technion
kfirylevy@technion.ac.il

**Ido Hakimi**
Technion
idohakimi@gmail.com

**Mark Silberstein**
Technion
mark@ee.technion.ac.il

## Abstract

We propose a novel technique for faster deep neural network training which systematically applies sample-based approximation to the constituent tensor operations, i.e., matrix multiplications and convolutions. We introduce new sampling techniques, study their theoretical properties, and prove that they provide the same convergence guarantees when applied to SGD training. We apply approximate tensor operations to single and multi-node training of MLP and CNN networks on MNIST, CIFAR-10 and ImageNet datasets. We demonstrate up to 66% reduction in the amount of computations and communication, and up to 1.37x faster training time while maintaining negligible or no impact on the final test accuracy.

## 1  Introduction

Approximation techniques for faster inference and training of deep neural networks (DNNs) have received considerable attention. Examples include quantization [1–5], low-rank and structured-sparse models [6–9], weight extrapolations [10], and partial/asynchronous gradient updates in the context of distributed training [11, 12]. Sampling-based approximations were used to accelerate inference [13, 14], but using them in training [15–17] has not been systematically studied nor demonstrated end-to-end GPU performance benefits in practice.

We propose a novel approach to accelerating *DNN training* by systematically approximating tensor operations via sampling. At a high level, the original matrix products and convolutions are replaced with their faster approximate versions. The approximation is applied separately to each tensor operation, keeping the network architecture and tensor dimensions intact, thereby facilitating the adoption of this technique in existing DNN training frameworks, potentially in combination with other approximation techniques. Furthermore, when combined with distributed training, our technique allows for seamless reduction in the communication bandwidth and increased performance gains.

We begin by reviewing the plethora of existing methods for approximating matrix multiplication. We compare several known algorithms [18–25], and find *column-row sampling (CRS)* [20] to be the most suitable for approximating matrix multiplications in training. In order to compute the product of two matrices $A^\top B$, the CRS algorithm samples the columns of $A^\top$ and the corresponding rows of $B$ thus constructing smaller matrices which are then multiplied as usual. This method incurs low sampling overheads and lends itself to an efficient implementation using existing dense matrix product routines. CRS minimizes the approximation error for the Frobenius norm of the resulting matrix while keeping the approximation unbiased.

---

*A Viterbi fellow

35th Conference on Neural Information Processing Systems (NeurIPS 2021).

Sampling-based approximations can be interpreted as a form of Dropout [26], and we discuss the similarities and differences between the two. While Dropout aims to prevent overfitting, we focus on approximations as means to accelerate training by reducing the amount of computation.

In this work we aim to answer two main questions. First, can neural networks be trained while using approximate tensor operations? Second, what are the relations between using exact or approximate operations during training?

We start by analyzing the simpler case of linear regression, where we can derive the effects of approximations in closed form. We define a new loss function that takes the sampling into account, and observe that the resulting gradients differ from the exact training due to the dependency between sampled features. To this end, we propose a new *Bernoulli-CRS* variant which achieves statistical independence of samples, study its properties, and show that in linear regression it is equivalent to dynamic $L_2$ weight regularization of the original, non-approximate loss.

We then turn to the more general case of non-linear deep neural networks. We show that using sampling-based approximations in the backward pass provides the same convergence guarantees as the exact SGD for bounded weights. The convergence result holds for unbounded weights as well if approximation is applied only to the weight gradients and if the activation functions are bounded.

We also study a new *TopK-CRS* algorithm which deterministically selects the top-$k$ column-row pairs with the highest norms. We show that this algorithm is equivalent to the minimal mean square error estimator (MMSE) in case column-row pairs are pairwise independent with zero mean.

Last, we generalize matrix product approximation to convolutions and analyze the approximation error to derive the optimal sampling policy. This allows us to apply approximations to training of convolutional neural networks (CNNs).

We implement our techniques in PyTorch [27] [2] and evaluate them on several DNN topologies, including MLP and CNN networks on MNIST [28], Wide ResNet 28-10 [29] on CIFAR-10 [30], and ResNet-50 and ResNet-152 [31] on ImageNet [32]. We demonstrate up to 66% reduction in the number of FLOPs and up to 1.33x faster training time with little or no degradation in model accuracy.

We develop another flavor of *TopK-CRS* which samples according to the weight norms only. When sampling the same subset of weights for different workers in a data-parallel setting, our sampling technique enables reducing the amount of gradient communication between workers. Notably, our algorithm is compatible with the standard AllReduce approach used in distributed deep learning. We implement an AllReduce scheme that takes advantage of the smaller gradient footprint and demonstrate 1.09x-1.37x speedup in multi-node training.

Our contributions are as follows:

- We derive general convergence guarantees for training with approximate tensor operations.
- We develop novel sampling algorithms and analyze their theoretical properties.
- We extend sampling-based algorithms to fast approximation of multi-channel convolutions.
- We show that our approach can reduce the computation, communication and total training time on several popular neural network architectures with little or no accuracy degradation.

## 2 Related work

To the best of our knowledge, we are the first to study the application of sample-based approximations of tensor operations to speed up DNN training. However, there have been several prior efforts to accelerate DNN computations via approximation which we survey below.

Several works accelerate inference through model compression [33–39]. A large body of work is devoted to quantization and low-precision datatypes (see for example [1, 2, 5]). Approximation was used to extrapolate weight values [10]. Another approach enforces low-rank or structured-sparse structure on the layers, resulting in lower computational cost both for training and inference [6–9]. Other works accelerate inference by approximating large matrices as products of lower-ranked ones [40, 41] or through locality-sensitive hashing [42].

---

[2]`https://github.com/acsl-technion/approx`

In the context of distributed training, several works targeted communication bottlenecks by gradient quantization [3, 4], delayed weight updates [11, 12] and low-rank approximation of the gradient matrix [43]. These methods are complementary and compatible with ours.

Sampling-based approximations were used to accelerate inference [13, 14], but using them for training[15–17] has not been systematically studied nor shown to speed up training on GPUs without accuracy degradation. Sub-sampling whole layers was shown to enable training of very deep CNNs [44].

## 2.1 Approximate matrix multiplication

There are several known algorithms for approximating matrix product. However, only those that meet the following requirements will be effective for DNN training. First, the algorithm should apply to dense matrices of arbitrary dimensions. Second, to reduce training time, the overall multiplication including input transformation should be faster than the original product. Last, the algorithm should be amenable to efficient implementation on commodity hardware.

Using these criteria, we consider the following algorithms:

**Random walk [18]** This algorithm performs random walks on a graph representation of the input matrices, but is applicable to non-negative matrices only.

**Random projections [21–23]** The two matrices to be multiplied are first projected into a lower-dimensional subspace by a scaled random sign matrix. These algorithms require both input matrices to be roughly square, otherwise the cost of projection will be similar to the cost of original product. In DNNs, however, it is common for one dimension to be smaller than the other.

**FFT [24, 25]** These algorithms represent each column-row outer product as a polynomial multiplication and then calculate it using Fast Fourier Transform. The complexity depends on the sparsity of the input matrices, decreasing as the sparsity increases. Therefore, these algorithms might not be effective for dense matrices.

**SVD [19, 33, 36]** Several algorithms replace one input matrix with its low-rank approximation using truncated SVD. These algorithms are suitable for inference where the weight matrix factorization can be pre-computed offline, but are not applicable to training since the high cost of factorization is incurred in every matrix product.

**Column-row sampling (CRS) [19, 20]** The sampling algorithm approximates matrix product $A^\top B$ by sampling $k$ columns of $A^\top$ and respective rows of $B$ to form smaller matrices, which are then multiplied as usual.

We choose CRS as the basis for our current work because it meets all the criteria above: It is applicable to fully-connected layers of any size, its effectiveness does not depend on the matrix contents, its sampling is computationally lightweight, and may use regular matrix multiplication algorithms since the sampled sub-matrices remain dense.

## 2.2 CRS

Let $A \in \mathbb{R}^{n \times m}, B \in \mathbb{R}^{n \times p}$. Their product $A^\top B$ is approximated as a weighted sum of outer products between sampled columns of $A^\top$ and corresponding rows of $B$:

$$A^\top B \approx \sum_{t=1}^{k} \frac{1}{k p_{i_t}} A^{\top (i_t)} B_{(i_t)} \tag{1}$$

where $A^{\top (i)}, B_{(i)}$ denote the matrix $i$'th column and row respectively, $k$ is the number of samples (satisfying $1 \le k \le n$), $\{p_i\}_{i=1}^{n}$ is a probability distribution over the column-row pairs of $A^\top, B$ and $i_t \in \{1, ..., n\}$. This algorithm allows linear reduction in complexity from $O(mnp)$ to $O(mkp)$.

(1) can also be expressed as $A^\top D S^\top S D B$, where $D \in \mathbb{R}^{n \times n}$ is a diagonal scaling matrix with:

$$(D)_{j,j} = \frac{1}{\sqrt{k p_j}} \tag{2}$$

and $S \in \mathbb{R}^{k \times n}$ is a sampling matrix that selects $k$ features, possibly with replacement. $S$ is a random matrix, where each row has 1 in one entry and zeros in others. In each row, the probability of having the non-zero entry in column $j$ is $p_j$.

Drineas et al. [20] show that CRS is unbiased:

$$\mathbb{E}\left[A^\top D S^\top S D B\right] = A^\top B \tag{3}$$

and also derive upper bounds for the expected Frobenius and spectral norms of the error matrix $\left\|A^\top B - A^\top D S^\top S D B\right\|$. They show that the error is minimized when the sampling probabilities are proportional to the product of the column-row Euclidean norms:

$$p_i = \frac{|A_{(i)}||B_{(i)}|}{\sum_{j=1}^n |A_{(j)}||B_{(j)}|} \tag{4}$$

In which case the expected Frobenius error is:

$$\frac{1}{k}\left(\sum_{t=1}^k |A_{(i_t)}||B_{(i_t)}|\right)^2 - \frac{1}{k}\left\|A^\top B\right\|_F^2 \tag{5}$$

### 2.3  Approximate Tensor Operations and Dropout

Sampling-based approximations can be interpreted as a flavor of Dropout [26], a popular technique to prevent overfitting by randomly zeroing individual activations during training. However, the sparsity pattern resulting from Dropout is unstructured and therefore cannot be exploited efficiently by GPUs despite recent advances in structured sparsity support[8]. Prior works on fast Dropout training [45, 46] are different than ours and do not demonstrate acceleration of large networks while maintaining accuracy.

Some works proposed non-uniform Dropout probabilities for individual activations [47] or full channels [48]. Their sampling heuristics are different from ours which are derived from optimal approximation. Furthermore, they use Dropout only for preventing overfitting and do not leverage it to speed up training. In our experiments we demonstrate the utility of sampling-based approximations for DNNs with and without Dropout. Conversely, we did not observe improved accuracy from approximations which could have been attributed to reduced overfitting.

## 3  Approximate Linear Regression

We now analyze CRS in the simpler context of linear regression, where we can derive the effects of approximations in closed form. We show that this leads to biased gradient estimates.

Let $X \subset \mathbb{R}^{n \times M}$ a dataset containing $M$ examples, each a vector in $\mathbb{R}^n$. Every $x^i \in X$ is associated with a "ground truth" value $y^i \in \mathbb{R}$.

Let $w \in \mathbb{R}^n$ be parameters in a linear model that predicts a value $\bar{y}^i \in \mathbb{R}$ for every $x^i \in X$:

$$\bar{y}^i = w^\top x^i \tag{6}$$

To simplify the notation we do not include an explicit bias term. We do so without loss of generality since we can always add another entry of 1 to the features.

Let us define the MSE (Mean Square Error) loss function:

$$\ell = \sum_{i=1}^M (\bar{y}^i - y^i)^2 \tag{7}$$

When using SGD (Stochastic Gradient Descent), we are given a single example $x^i \in \mathbb{R}^n$ in each step, and update $w$ using the gradients $\frac{\partial \ell}{\partial w}$. For notation simplicity we omit the superscript $i$ from $x^i, \bar{y}^i, y^i$.

The gradients are given by the chain rule as:

$$\frac{\partial \ell}{\partial w} = 2x(w^\top x - y) \tag{8}$$

Now, let us assume that the multiplication $w^\top x$ is approximated using CRS. The linear regression model now becomes:

$$\hat{y} = w^\top D S^\top S D x \tag{9}$$

and for the MSE loss the gradients will be:

$$\widehat{\frac{\partial \ell}{\partial w}} = 2 D S^\top S D x (w^\top D S^\top S D x - y) \tag{10}$$

Where $\widehat{\frac{\partial \ell}{\partial w}}$ denotes the CRS weight gradients.

Note that $D S^\top S D$ appears twice in (10). In $w^\top D S^\top S D x$ it represents sampling in the forward pass, while in $D S^\top S D x$ it results in passing gradients only to the elements of $w$ that were sampled in the forward pass.

It should be emphasized that (9) and (10) in fact describe gradients with respect to a different loss function compared to (8): one loss function uses $\hat{y}$ while the other uses $\bar{y}$. If the approximate gradients are unbiased estimates of the non-approximate gradients, we could relate the approximate training process to the original one. However, the weight gradients do not satisfy this unbiasedness property:

$$\mathbb{E}\left[(\widehat{\frac{\partial \ell}{\partial w}})_j\right] = 2x_j \left(\sum_{t=1}^n w_t \mathbb{E}\left[(\tilde{S})_{j,j}(\tilde{S})_{t,t}\right] x_t - y\right) \tag{11}$$

where we denote $\tilde{S} \triangleq D S^\top S D$, and use the fact that $\tilde{S}$ is diagonal.

For the expression in (11) to be equal to that in (8) we need that $\mathbb{E}\left[(\tilde{S})_{j,j}(\tilde{S})_{t,t}\right] = 1$. However, this is not the case because $(\tilde{S})_{j,j}$ and $(\tilde{S})_{t,t}$ are not independent random variables: an entry in the diagonal of $\tilde{S}$ is the (scaled) number of times a column-row pair was selected out of the $k$ total samples in CRS. The event of selecting a particular pair therefore affects selecting others.

We note that if instead of changing the loss function we treated the approximate multiplication as a "black box" that replaces the original product, we could use (8) and only replace the forward pass product $w^\top x$ with the "black box" substitute of $w^\top D S^\top S D x$. This would yield:

$$\widetilde{\frac{\partial \ell}{\partial w}} = 2x(w^\top D S^\top S D x - y) \tag{12}$$

which satisfies:

$$\mathbb{E}\left[\widetilde{\frac{\partial \ell}{\partial w}}\right] = \frac{\partial \ell}{\partial w}. \tag{13}$$

(12) is equivalent to applying the approximate computation in the forward pass, but propagating the gradients to all weight entries in the same way as if the computation were exact. In practice we find that this approach leads to significantly lower accuracy in deep neural networks compared to sampling the same entries in the forward and backward pass, or to applying approximations in the backward pass only.

## 4 Bernoulli-CRS

We now turn to develop Bernoulli-CRS, a new variant of sampling approximation that enables to sample column-row pairs independently and without replacement. Applied to linear regression, we show that using Bernoulli-CRS is equivalent to employing unbiased gradient estimates in addition to a bias term which can be interpreted as scale dependent weight regularization.

**Bernoulli-CRS:** These aforementioned properties can be achieved by assigning a separate Bernoulli sampling probability $p_i$ for each column-row pair $i$, and sampling pairs independently of each other. To control the amount of sampling, we add another constraint that all the probabilities will sum up to an integer $k$:

$$\sum_{i=1}^{n} p_i = k \tag{14}$$

Let us define $K \in \mathbb{R}^{n \times n}$ a random diagonal sampling matrix where $K_{j,j} \sim \text{Bernoulli}(p_j)$ for $1 \leq j \leq n$. Furthermore, let us define another diagonal scaling matrix $P \in \mathbb{R}^{n \times n}$ where $P_{j,j} = \frac{1}{\sqrt{p_j}}$ for $1 \leq j \leq n$.

Using the $K$ and $P$ matrices we may now define our new Bernoulli-CRS algorithm. Let $A \in \mathbb{R}^{n \times m}$ and $B \in \mathbb{R}^{n \times p}$. The product $A^\top B$ can be approximated with $\tilde{A}^\top \tilde{B}$ defined as follows:

$$\tilde{A}^\top \tilde{B} := \sum_{i=1}^{n} \frac{Z_i}{p_i} A^{\top(i)} B_{(i)} = A^\top P K K P B \tag{15}$$

where $\{Z_i \sim \text{Bernoulli}(p_i)\}_{i=1}^{n}$ are independent random variables. We denote $\tilde{A} \triangleq KPA$ and $\tilde{B} \triangleq KPB$.

In the appendix we develop the properties of Bernoulli-CRS. We show it is unbiased and derive bounds on the error variance both in expectation and in high probability. We derive the optimal sampling probabilities minimizing the expected variance, and show that under certain conditions they are given by the simpler expression:

$$p_i = \min \left\{ \frac{k|A_{(i)}||B_{(i)}|}{\sum_{j=1}^{n} |A_{(j)}||B_{(j)}|}, 1 \right\} \tag{16}$$

In the appendix we show that applying Bernoulli-CRS in linear regression leads to unbiased estimate of the original gradients with an additional regularization term $\mathcal{R}(w)$, which we define as:

$$\mathcal{R}(w) = \mathbf{E} \left[ \sum_{j=1}^{n} \frac{1 - p_j}{p_j} x_j^2 w_j^2 \right] \tag{17}$$

and the expectation is with respect to the distribution of the data samples.

The term $\mathcal{R}(w)$ can be interpreted as input-dependent $L_2$ regularization. The regularization is higher as $x_j$ grows in magnitude and as $p_j$ decreases. Both serve to reduce the impact on the weights if they were chosen with small probabilities or mostly because of the input size. We note that Wager et al. [49] conduct a similar analysis for Dropout in the particular case where the same sampling probability is used for all features.

To summarize, sampling in the simpler case of linear regression minimizes the original loss function with an added regularization term.

## 5 Approximate Backpropagation in Non-Linear Deep Networks

The analysis of approximate linear regression cannot simply generalize to deep non-linear networks: non-linearity leads to biased network output even if the approximate multiplication is itself unbiased. Still, we are able to obtain strong theoretical results on the relations between exact and approximate training if the approximations are limited to the backward pass: the forward pass is calculated as usual, and the matrix products in the backward pass are performed using approximate matrix multiplication.

We prove the following theorem:

**Theorem 1.** *Let $f(x, W, b)$ be a multi-layer neural network with $\beta$-Lipschitz activation functions $\sigma$. Let $\ell$ be a $\beta$-Lipschitz loss function, and let the network be trained with SGD using properly*

*decreasing learning rate. Assume that the weights are bounded; and further assume that the matrix products in the backward pass are approximated using an unbiased approximation scheme, i.e.,*

$$\mathbb{E}\left[A^\top B - \texttt{approx}(A^\top B)\right] = 0$$

*and that there exists a constant $C$ and a norm $||\cdot||$ such that:*

$$\mathbb{E}\left[\left|\left|A^\top B - \texttt{approx}(A^\top B)\right|\right|^2\right] \le C\,||A||^2\,||B||^2.$$

*Then the approximated NN gradient estimates are unbiased, and their second moments are bounded.*

**Corollary.** *Based on recent works on non-convex optimization (see e.g. [50]), the unbiasedness and bounded second moments ensured by Theorem 1 imply that approximate backpropagation enjoys the same convergence guarantees as regular SGD training.*

In the appendix we show that CRS and other sampling algorithms satisfy the property

$$\mathbb{E}\left[\left|\left|A^\top B - \texttt{approx}(A^\top B)\right|\right|^2\right] \le C\,||A||^2\,||B||^2$$

Note that for Theorem 1 we required that weights will be bounded during the training process. This is a strong assumption which could be justified if weight regularization or clipping is used. In the appendix we prove the same results without relying on these assumptions, if only the weight gradients are approximated and if the activation function is bounded (such as sigmoid).

## 6 Sampling Without Scaling and Top-$k$ Selection

We now consider a different sampling scheme where $k$ column-row pairs are selected deterministically without scaling. This can be viewed as a special case of Bernoulli-CRS, where the sampling probabilities are either 0 or 1. We now show that under certain assumptions on the distribution of the input matrices, this scheme can lead to the optimal estimation:

**Theorem 2.** *Let $A$ be an $n \times m$ random matrix and $B$ be an $n \times p$ random matrix, such that*

$$\mathbb{E}\left[A^{\top(i)}B_{(i)}\right] = 0$$

*for $1 \le i \le n$. Assume $k$ column-row pairs with indices $\{j\}_1^n$ are sampled from $A$ and $B$. Then, the MMSE estimator of the product $A^\top B$ is $\tilde{A}^\top \tilde{B}$ where $\tilde{A}, \tilde{B}$ are constructed from the sampled column-row pairs without scaling.*

*Furthermore, if $A^{\top(i)}B_{(i)}$ and $A^{\top(j)}B_{(j)}$ are independent for different $i, j$ then the MSE is minimized when sampling $k$ pairs with the maximum norm multiplication $|A_{(i)}||B_{(i)}|$.*

The assumptions in Theorem 2 can hold in practice if weights are initialized with a zero-centered distribution[51], if the distribution of weights remains centered around zero during training [51–54], and if different weights can be considered pairwise-independent [55].

We study the approximation quality of CRS, Bernoulli-CRS and top-$k$ selection on synthetic matrix multiplication. We generate $100 \times 100$ random matrices and compute the error metric:

$$\frac{\left|\left|A^\top B - \texttt{approx}(A^\top B)\right|\right|_F}{||A||_F\,||B||_F} \tag{18}$$

Figures 1(a),1(b) show the approximation error for different algorithms and sampling ratios, averaged over 1000 runs. We observe that Bernoulli-CRS outperforms CRS in higher sampling ratios. Also, when one matrix has i.i.d entries with zero mean, Bernoulli-CRS outperforms CRS and top-$k$ selection performs the best as expected from Theorem 2.

We also consider a different flavor of top-$k$ selection, which we refer to as "top-$k$-weights": sampling $k$ column-row pairs corresponding to rows of $B$ with the highest norms. While not providing the theoretical guarantees of Theorem 2, the new variant has a desirable property for data parallel distributed training, where weights are identical between different workers. A deterministic selection algorithm that only depends on the weights will sample the same weights for all workers, allowing to reduce the gradient communication between nodes to the selected weights only.

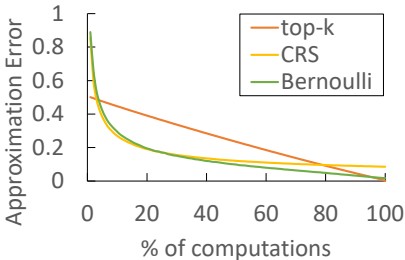
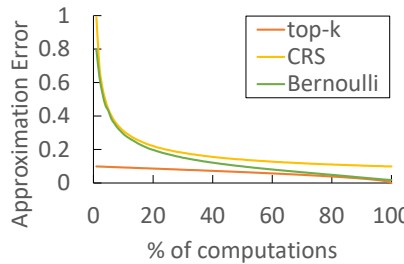

(a) Matrix product: both matrix entries drawn from $\mathcal{N}(1,1)$

(b) Matrix product: one matrix entries drawn from $\mathcal{N}(0,1)$, the other from $\mathcal{N}(1,1)$

Figure 1: Approximation error depending on the amount of performed computations. Lower is better.

# 7 Approximating Convolutions

We extend the basic CRS algithm to the approximation of multi-channel convolutions. In matrix multiplication sampling is performed over the common dimension. The analogue for multi-channel convolution is to sample over the input channels dimension, illustrated in Figure 2. As in the matrix case, the output dimensions remain the same.

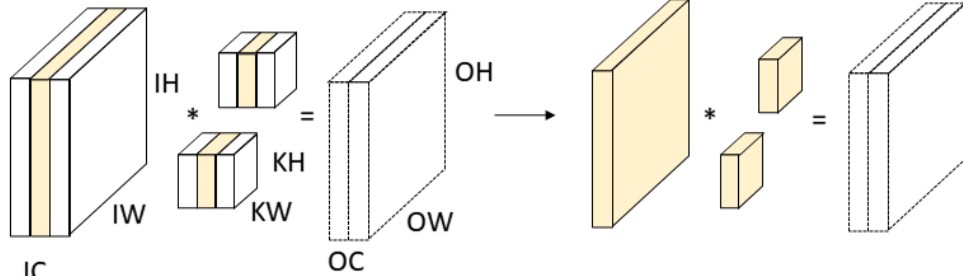

Figure 2: Sampling one input channel out of three

In the appendix we derive the optimal sampling probabilities and scaling factors. Bernoulli-CRS and top-$k$ algorithms can be developed for convolutions as well in an analogous way.

# 8 Experimental Results

We implement CRS, Bernoulli-CRS and top-$k$ selection approximation algorithms in PyTorch both for matrix multiplication and convolution. Our implementation allows to control the sampling degree and the application of approximation in the forward or backward passes.

We replace exact tensor operations with their sampling-based approximations, *without changing training hyper-parameters*. Only column-row pairs sampled in the forward pass are used during backpropagation as the rest do not affect the loss. Hence, sampling in the forward pass also reduces the amount of backward pass computations by the same ratio. We apply approximations only during training, and use exact computations for validation/test evaluation.

We evaluate our approximate training technique on several network architectures and datasets: MLP and CNN on MNIST, Wide ResNet 28-10 on CIFAR-10 and ResNet-50 and ResNet-152 on ImageNet. We train the networks on a single node using NVidia V100 GPUs (two GPUs for ResNet-152, one for the rest), and measure the reduction in multiply-accumulate operations due to sampling as well as the overall speedup in total training time versus the non-approximate baseline. The appendix includes additional details on the models and the training process.

Table 1: Compute reduction, communication reduction and wall-clock speedup of training with approximate tensor operations.

| NETWORK | | COMPUTE REDUCTION | COMMUNICATION REDUCTION | ACCURACY (BASELINE) | TRAINING SPEEDUP |
|---|---|---|---|---|---|
| MLP (MNIST) | | 50% | - | 98.22% (98.22%) | - |
| CNN (MNIST) | | 66% | - | 99.25% (99.35%) | - |
| WRN-28-10 (CIFAR-10) | | 50% | - | 95.89% (96.17%) | 1.33x |
| RESNET-50 (IMAGENET) | | 6.5% | - | 75.63% (75.6%) | 1.04x |
| RESNET-152 (IMAGENET) | SINGLE NODE | 40% | - | 76.44% (77.65%) | 1.16x |
| | | 9% | - | 77.66% (77.65%) | 1.04x |
| | 8 NODES | 40% | 48% | 76.44% (77.65%) | 1.37x |
| | | 12% | 23% | 77.48% (77.65%) | 1.13x |
| | | 9% | 13% | 77.8% (77.65%) | 1.09x |

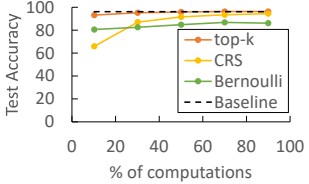

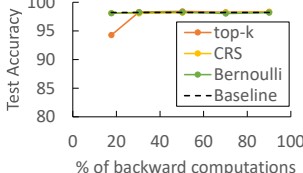

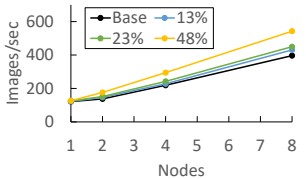

Figure 3: WRN-28-10 on CIFAR-10 (approximate forward and backward)

Figure 4: 3-layer MLP on MNIST (exact forward, approximate backward)

Figure 5: AllReduce with top-k-weights sampling (% fewer gradients sent).

Our results using top-$k$ sampling are summarized in Table 1. We see a reduction of up to 66% in the computational cost with little or no degradation in model accuracy, and up to 1.33x faster training time. We believe the gap between compute reduction and end-to-end speedup can be reduced by fusing the sampling with the matrix multiplication routines, or running on a different HW architecture that allows fast sampling and norm computation. We note that the small MNIST models do not exhibit training time speedup since they are not compute-intensive enough to saturate the GPU. The ratio between compute reduction and actual speedup is smaller in ResNet-152 compared to ResNet-50 and WRN-28-10 because the batch size per GPU is lower due to the limited GPU memory capacity.

**Sampling Algorithms** We compare CRS, Bernoulli-CRS and top-$k$ selection on MNIST and CIFAR-10 and find empirically that top-$k$ results in higher accuracy and faster training time (Figure 3). This result is consistent with that of approximate $\mathcal{N}(0, 1)$ matrix product (Figure 1(b)). This is not surprising given Theorem 2 and our empirical observation that the weight distribution is close to symmetrical around zero throughout training.

**Approximations in Forward Pass and Backpropagation** For the small MNIST models we are able to perform as low as 10% of the computations in the backward pass without harming accuracy (Fig. 4). However, in the larger models (WRN-28-10) we find empirically that accuracy drops when approximating only the backward pass. Therefore, in Table 1 we report results for consistent sampling in the forward and backward passes.

**Sampling Ratio Sensitivity** We find that the achievable compute reduction is not consistent across networks and datasets. For MNIST and CIFAR-10 we maintain good accuracy while reducing 50%-66% of the computations. However, ImageNet proved to be more sensitive and we kept the accuracy intact when applying 50% sampling to the ResNet layers with 1024 or more channels only. Figure 6 shows the learning curves under different sampling ratios compared to the non-approximate baseline.

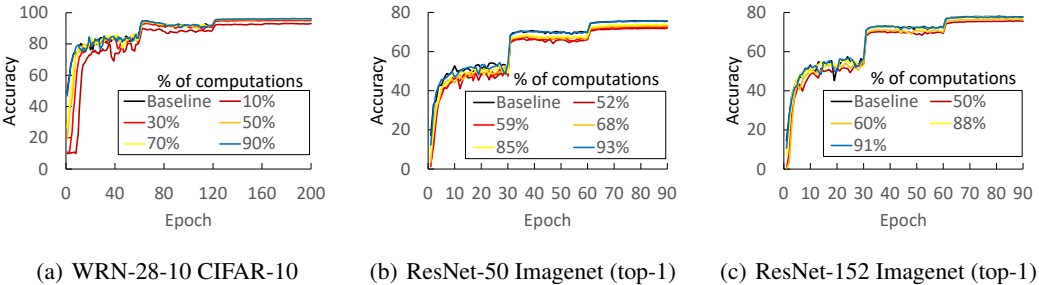

| (a) WRN-28-10 CIFAR-10 | (b) ResNet-50 Imagenet (top-1) | (c) ResNet-152 Imagenet (top-1) |

Figure 6: Learning curves for validation accuracy under different top-$k$ sampling ratios

**Distributed Training** We implement an AllReduce "top-$k$-weights" scheme in PyTorch. This scheme performs reduction only for the gradients of the sampled weights, reducing inter-node communications. Table 1 shows the accuracy-speedup trade-off for ResNet-152 distributed training. Figure 5 shows the respective scaling behavior of these schemes relative to the exact baseline. We note that compute savings did not lead to significant single-node speedup since in this experiment the V100 GPUs (from Amazon AWS) had lower memory capacity, which led to smaller batch size per GPU. The multi-node training speedup is therefore mostly due to the communication savings.

## 9 Conclusion

In this work we have demonstrated the utility of sample-based approximation of tensor operations for neural network training, both theoretically and empirically. We believe that further acceleration could be achieved through dedicated GPU primitives fusing sampling and matrix multiplication/convolution, as well as varying and adaptive sampling rates for different layers and iterations. Studying other approximation algorithms, applications in resource-constrained environments and bridging the gaps between our theoretical results and what worked best in practice are all promising directions for future research. Overall, we believe that sample-based approximations and fast approximations in general are valuable additions to the toolbox of techniques for deep learning acceleration.

## Acknowledgments and Disclosure of Funding

K.Y. Levy acknowledges support from the Israel Science Foundation (grant No. 447/20). I. Hakimi acknowledges support from the Hasso Plattner Institute at the Technion. M. Silberstein acknowledges support from the Israel Science Foundation (grant No. 1027/18).

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
