# Appendices

## A    Bernoulli-CRS Properties

Let us define $K \in \mathbb{R}^{n \times n}$ a random diagonal sampling matrix where $K_{j,j} \sim \text{Bernoulli}(p_j)$ for $1 \leq j \leq n$.

Let us define another diagonal scaling matrix $P \in \mathbb{R}^{n \times n}$ where $P_{j,j} = \frac{1}{\sqrt{p_j}}$ for $1 \leq j \leq n$.

Using the $K$ and $P$ matrices we may now define our new Bernoulli-CRS algorithm. Let $A \in \mathbb{R}^{n \times m}$ and $B \in \mathbb{R}^{n \times p}$. The product $A^\top B$ can be approximated with $\tilde{A}^\top \tilde{B}$ defined as follows:

$$\tilde{A}^\top \tilde{B} := \sum_{i=1}^{k} \frac{Z_i}{p_i} A^{\top (i)} B_{(i)} = A^\top P K K P B \tag{19}$$

where $\{Z_i \sim \text{Bernoulli}(p_i)\}_{i=1}^n$ are independent random variables. We denote $\tilde{A} \triangleq KPA$ and $\tilde{B} \triangleq KPB$.

First, we show that the above holds in expectation:

**Proposition 1.** $\mathbb{E}\left[\tilde{A}^\top \tilde{B}\right] = A^\top B$.

Let $T = \text{Trace}(K)$ the number of non-zero diagonal elements in $K$. We note that to perform the actual computation it is enough to sample the $T$ column-row pair with the corresponding element in $K$ being non-zero. Unlike CRS, the lower rank of the sampled matrices is not constant and depends on the random matrix $K$. Its expectation is controlled through the parameter $k$:

**Proposition 2.** $\mathbb{E}[T] = k$.

Therefore, Bernoulli-CRS will perform on average the same amount of computations as in the fixed-rank CRS.

Let us further derive the properties of the proposed sampling algorithm. Specifically, what are the optimal values for the probabilities $p_i$ under the constraint $\sum_{i=1}^{n} p_i = k$?

First, let us calculate the variance of $\tilde{A}^\top \tilde{B}$:

**Proposition 3.**

$$\mathbf{Var}\left[(\tilde{A}^\top \tilde{B})_{i,j}\right] = \sum_{t=1}^{n} \frac{1 - p_t}{p_t} A_{t,i}^2 B_{t,j}^2$$

We will be interested in the Frobenius norm of the error matrix $\left|\left|A^\top B - \tilde{A}^\top \tilde{B}\right|\right|_F^2$, which can be derived from the following theorem:

**Theorem 3.** *The expected Frobenius norm of the error matrix* $\mathbb{E}\left[\left|\left|A^\top B - \tilde{A}^\top \tilde{B}\right|\right|_F^2\right]$ *is* $\sum_{t=1}^{n} \frac{1 - p_t}{p_t} |A_{(t)}|^2 |B_{(t)}|^2$.

*Furthermore, under the constraint* $\sum_{i=1}^{n} p_i = k$ *it is minimized for the probabilities:*

$$p_i = \frac{|A_{(i)}||B_{(i)}|}{\sqrt{\mu}} 1_{\{0 < |A_{(i)}||B_{(i)}| < \mu\}} + 1_{\{|A_{(i)}||B_{(i)}| \geq \mu\}}$$

*where $\mu$ is a root of the following function:*

$$G(\mu) := \sum_{i=1}^{n} \left(\frac{|A_{(i)}||B_{(i)}|}{\sqrt{\mu}} 1_{\{0 < |A_{(i)}||B_{(i)}| < \mu\}} + 1_{\{|A_{(i)}||B_{(i)}| \geq \mu\}}\right) - k$$

**Corollary.** *The sampling probabilities*

$$p_i = \min \left\{ \frac{k|A_{(i)}||B_{(i)}|}{\sum_{j=1}^n |A_{(j)}||B_{(j)}|}, 1 \right\}$$

*are optimal if $k \leq \frac{\sum_{i=1}^n |A_{(i)}||B_{(i)}|}{\max_i |A_{(i)}||B_{(i)}|}$*

From Theorem 3 it follows that for the probabilities:

$$p_i = \min \left\{ \frac{k|A_{(i)}||B_{(i)}|}{\sum_{j=1}^n |A_{(j)}||B_{(j)}|}, 1 \right\} \tag{20}$$

the expected Frobenius error is:

$$\frac{1}{k} \left( \sum_{i=1}^n e_i |A_{(i)}||B_{(i)}| \right)^2 - \sum_{i=1}^n e_i |A_{(i)}|^2 |B_{(i)}|^2 \tag{21}$$

where we denote:

$$e_i \triangleq \begin{cases} 1 & |A_{(i)}||B_{(i)}| \leq \frac{\sum_{j=1}^n |A_{(j)}||B_{(j)}|}{k} \\ 0 & \text{else} \end{cases}. \tag{22}$$

Comparing that with the bound in (5), we can see that different values of $A, B$ determine which algorithm performs better.

Knowing the expected Frobenius error also implies a bound on the spectral norm of the error matrix, since the spectral and Frobenius norms are related by:

$$||A|| \leq ||A||_F \leq \sqrt{r} \, ||A|| \tag{23}$$

where $r$ is the rank of $A$ and $||A||$ denotes its spectral norm.

The following theorem yields high probability bounds for the Frobenius and spectral norms for the Bernoulli-CRS algorithm:

**Theorem 4.** *Let $A \in \mathbb{R}^{n \times m}$ and $B \in \mathbb{R}^{n \times p}$. Let $\tilde{A}, \tilde{B}$ be the sampled matrices according to the Bernoulli-CRS algorithm described above. Denote*

$$R \triangleq \max_i \left\| A^{\top (i)} B_{(i)} \right\|$$

*and*

$$\sigma^2 \triangleq \frac{1}{k} \left( \sum_{i=1}^n e_i |A_{(i)}||B_{(i)}| \right)^2 - \sum_{i=1}^n e_i |A_{(i)}|^2 |B_{(i)}|^2$$

*then, for all $t \geq 0$:*

$$\mathbb{P} \left\{ \left\| A^\top B - \tilde{A}^\top \tilde{B} \right\| \geq t \right\} \leq (m+p) \cdot \exp \left( \frac{-t^2/2}{\sigma^2 + Rt/3} \right)$$

$$\mathbb{P} \left\{ \left\| A^\top B - \tilde{A}^\top \tilde{B} \right\|_F \geq t \right\} \leq (m+p)^{3/2} \cdot \exp \left( \frac{-t^2/2}{\sigma^2 + Rt/3} \right)$$

### A.1 Bernoulli-CRS in linear regression

We now show that applying Bernoulli-CRS in linear regression leads to unbiased estimate of the original gradients with an additional term that can be interpreted as regularization. The analysis for linear regression using Bernoulli-CRS is the same as in Section 3, with the sampling and scaling matrices $DS^\top SD$ replaced with $PKKP$

The expression for the weight gradient (simimlar to (11)) now becomes:

$$\mathbb{E}\left[(\widehat{\frac{\partial \ell}{\partial w}})_j\right] = 2x_j(\sum_{t=1}^{n} w_t(\mathbb{E}\left[(\tilde{K})_{j,j}(\tilde{K})_{t,t}\right]x_t - y) \tag{24}$$

$$= 2x_j(w^\top x - y + w_j\left(\mathbb{E}\left[(\tilde{K})_{j,j}^2\right] - 1\right)x_j) \tag{25}$$

$$= 2x_j(w^\top x - y + \frac{1 - p_j}{p_j}w_j x_j) \tag{26}$$

where we denote $\tilde{K} \triangleq PKKP$.

When comparing (26) and (8) we see that using Bernoulli-CRS yields unbiased estimates of the original gradients with an additional bias term that is related to a scale-dependent regularization $\mathcal{R}(w)$, which we define as:

$$\mathcal{R}(w) = \mathbf{E}\left[\sum_{j=1}^{n} \frac{1 - p_j}{p_j}x_j^2 w_j^2\right] \tag{27}$$

and the expectation is with respect to the distribution of the data samples.

This term can be interpreted as input-dependent $L_2$ regularization. The regularization is higher as $x_j$ grows in magnitude and as $p_j$ decreases. Both serve to reduce the impact on the weights if they were chosen with small probabilities or mostly because of the input size.

## B  Approximating Convolutions - Details

Formally, let $I \in \mathbb{R}_{B \times IH}^{IW \times IC}$ be the input tensor, where $B$ is the batch size, $IH, IW$ are the input height and width, and $IC$ are the input channels. Let $K \in \mathbb{R}_{KH \times KW}^{IC \times OC}$ be the kernels tensor, where $KH, KW$ are the kernel height and width, and $IC, OC$ are the input and output channels respectively. Let $O \in \mathbb{R}_{B \times OH}^{OW \times OC}$ be the output tensor, where $OH, OW$ are the output height and width.

The multi-channel convolution operation is defined as:

$$O_{b,oh}^{ow,oc} = I * K = \sum_{i=1}^{IC} \sum_{kh=1}^{KH} \sum_{kw=1}^{KW} I_{b,oh+kh-1}^{ow+kw-1,i} \cdot K_{kh,kw}^{i,oc} \tag{28}$$

For notation simplicity, we assume zero padding. The inner sums in (28) can be written as 1-channel convolutions:

$$O_{b,oh}^{ow,oc} = \sum_{i=1}^{IC} I^{[i]} * K_{[i]} \tag{29}$$

where $I^{[i]} \in \mathbb{R}_{B \times IH}^{IW \times 1}$ denotes a tensor with one input channel that corresponds to the $i$'th input channel of $I$, i.e $I^{[i]}_{b,ih}^{iw,1} = I_{b,ih}^{iw,i}$. Similarly, $K_{[i]} \in \mathbb{R}_{KH \times KW}^{1 \times OC}$ corresponds to the $i$'th input channel of $K$.

This formulation immediately hints at the possibility to sample over the input channel dimension, similarly to sampling column-row pairs in matrices. We propose to approximate convolutions by sampling lower-rank tensors:

$$\tilde{O} = \sum_{t=1}^{k} \frac{1}{kp_{i_t}} I^{[i_t]} * K_{[i_t]} \triangleq \tilde{I} * \tilde{K} \tag{30}$$

where $\{i_t\}_{t=1}^{k}$ are such that $i_t \in \{1, ..., IC\}$ and $\{p_i\}_{i=1}^{IC}$ is a probability distribution over the input channels, $\tilde{I}$ is a tensor composed of sampled input channels of $I$ scaled by $\sqrt{\frac{1}{kp_i}}$, and $\tilde{K}$ is a tensor composed of corresponding sampled input channels of $K$ scaled by the same factor.

Computing the convolution of the smaller tensors $\tilde{I} * \tilde{K}$ can be done using standard efficient convolution implementations. Figure 2 illustrates the sampling operation.

The properties of the approximation in (30) can be derived similarly to the CRS derivations for matrix multiplication. In particular, we prove the approximation is unbiased, and similar to matrix CRS, we use sampling probabilities proportional to the tensor Euclidean norms:

$$p_i = \frac{\left\|I^{[i]}\right\|_F \cdot \left\|K_{[i]}\right\|_F}{\sum_{j=1}^{IC} \left\|I^{[j]}\right\|_F \cdot \left\|K_{[j]}\right\|_F} \tag{31}$$

In section C.4 we show that the optimal sampling probabilities are significantly more complicated to calculate, but under certain conditions they reduce to (31).

Bernoulli-CRS and top-$k$ algorithms can be developed for convolutions as well in an analogous way.

## C    Proofs

### C.1    Proofs for Section 5 - Approximate Backpropagation

**Theorem 1.** *Let $f(x, W, b)$ be a multi-layer neural network with $\beta$-Lipschitz activation functions $\sigma$. Let $\ell$ be a $\beta$-Lipschitz loss function, and let the network be trained with SGD using properly decreasing learning rate. If the matrix products in the backward pass are approximated using an unbiased approximation scheme satisfying:*

$$\mathbb{E}\left[A^\top B - \texttt{approx}(A^\top B)\right] = 0$$

*and:*

$$\mathbb{E}\left[\left|\left|A^\top B - \texttt{approx}(A^\top B)\right|\right|^2\right] \le C \left|\left|A\right|\right|^2 \left|\left|B\right|\right|^2$$

*for some finite constant $C$ and some norm $||\cdot||$,*

*and if the weights are bounded, then the approximated gradients are unbiased with bounded second moments.*

**Corollary.** *Based on recent works on non-convex optimization [50], Theorem 1 implies that approximate backpropagation enjoys the same convergence guarantees as regular SGD training.*

*Proof.* The network $f$ can be described by:

$$
\begin{aligned}
h_1 &= W_1^\top x + b_1 \\
a_1 &= \sigma(h_1) \\
h_l &= W_l^\top a_{l-1} + b_l \\
a_l &= \sigma(h_l) \\
\hat{y} &= W_L^\top a_{L-1}
\end{aligned}
$$

where $x \in \mathbb{R}^n$, $W_1 \in \mathbb{R}^{n \times d_1}$, $W_l \in \mathbb{R}^{d_{l-1} \times d_l}$, $b_l \in \mathbb{R}^{d_l}$, $\ell$ is the number of layers and $\hat{y} \in \mathbb{R}^{d_L}$ is the network output.

Let us denote the weight, bias and activation gradients with respect to a loss function $\ell$ by $\nabla W_l, \nabla b_l, \nabla a_l$ respectively. Let us denote and the gradients yielded by the approximation scheme as $\nabla \tilde{W}_l, \nabla \tilde{b}_l, \nabla \tilde{a}_l$.

**Lemma 1.**
$$\mathbb{E}\left[\nabla \tilde{W}_l\right] = \nabla W_l \quad and \quad \mathbb{E}\left[\nabla \tilde{b}_l\right] = \nabla b_l$$

*Proof.* We prove by induction. The last layer satisfies:

$$\nabla W_L = a_{L-1} \nabla \hat{y} \qquad \nabla a_{L-1} = W_L \nabla \hat{y}$$

and its approximation is given by:

$$\nabla \tilde{W}_L = \mathtt{approx}(a_{L-1}\nabla \hat{y}) \qquad \nabla \tilde{a}_{L-1} = \mathtt{approx}(W_L \nabla \hat{y})$$

Since the approximation methods satisfies:

$$\mathbb{E}\left[A^\top B - \mathtt{approx}(A^\top B)\right] = 0$$

we get:

$$\mathbb{E}\left[\nabla \tilde{W}_L\right] = \nabla W_L \qquad \mathbb{E}\left[\nabla \tilde{a}_{L-1}\right] = \nabla a_{L-1}$$

for the induction step, we will show that if $\mathbb{E}\left[\nabla \tilde{a}_l\right] = \nabla a_l$ then:

$$\mathbb{E}\left[\nabla \tilde{W}_{l-1}\right] = \nabla W_{l-1}$$

$$\mathbb{E}\left[\nabla \tilde{b}_{l-1}\right] = \nabla b_{l-1}$$

$$\mathbb{E}\left[\nabla \tilde{a}_{l-1}\right] = \nabla a_{l-1}$$

$\nabla \tilde{W}_{l-1}$ is given by:

$$\nabla \tilde{W}_{l-1} = \mathtt{approx}(a_{l-1}\nabla \tilde{h}_l^\top) = \mathtt{approx}(a_{l-1}\Sigma'(h_l)\nabla \tilde{a}_l^\top)$$

where $\Sigma'(h_l)$ is a diagonal matrix with the diagonal being the derivative of $\sigma$ in location $h_l$. Taking the expectation we get:

$$\begin{aligned}
\mathbb{E}\left[\nabla \tilde{W}_{l-1}\right] &= \mathbb{E}\left[\mathtt{approx}(a_{l-1}\Sigma'(h_l)\nabla \tilde{a}_l^\top)\right] \\
&= \mathbb{E}\left[\mathbb{E}\left[\mathtt{approx}(a_{l-1}\Sigma'(h_l)\nabla \tilde{a}_l^\top)|\nabla \tilde{a}_l^\top\right]\right] \\
&= \mathbb{E}\left[a_{l-1}\Sigma'(h_l)\nabla \tilde{a}_l^\top\right] \\
&= a_{l-1}\Sigma'(h_l)\nabla a_l^\top \\
&= \nabla W_{l=1}
\end{aligned}$$

where we used the unbiased approximation property of $\mathtt{approx}$ and the law of total expectation. Similar arguments for $\mathbb{E}\left[\nabla \tilde{a}_{l-1}\right]$ yield:

$$\begin{aligned}
\mathbb{E}\left[\nabla \tilde{a}_{l-1}\right] &= \mathbb{E}\left[\mathtt{approx}(W_l\nabla \tilde{h}_l)\right] \\
&= \mathbb{E}\left[\mathtt{approx}(W_l\Sigma'(h_l)\nabla \tilde{a}_l)\right] \\
&= \mathbb{E}\left[\mathbb{E}\left[\mathtt{approx}(W_l\Sigma'(h_l)\nabla \tilde{a}_l)|\nabla \tilde{a}_l\right]\right] \\
&= \mathbb{E}\left[W_l\Sigma'(h_l)\nabla \tilde{a}_l\right] \\
&= W_l\Sigma'(h_l)\nabla a_l \\
&= \nabla a_{l-1}
\end{aligned}$$

and for $\mathbb{E}\left[\nabla \tilde{b}_{l-1}\right]$:

$$\begin{aligned}
\mathbb{E}\left[\nabla \tilde{b}_{l-1}\right] &= \mathbb{E}\left[\nabla \tilde{h}_{l-1}\right] \\
&= \mathbb{E}\left[\Sigma'(h_l)\nabla \tilde{a}_l\right] \\
&= \Sigma'(h_l)\nabla a_l \\
&= \nabla b_{l-1}
\end{aligned}$$

$\square$

In other words, the unbiased estimation of the gradients follows from the linearity of backpropagation with respect to the gradients, even for non-linear activation functions.

We can write the training step using SGD and the approximate gradients $\nabla \tilde{W}_l^t$ for layer $l$ at iteration $t$ as:

$$W_l^{t+1} = W_l^t - \alpha_t(\nabla W_l^t + \omega_t)$$

where $\omega_t$ is a gradient noise defined as:

$$\omega_t \triangleq \nabla \tilde{W}_l^t - \nabla W_l^t$$

Based on Lemma 1, the gradient noise $\omega_t$ is a martingale difference sequence satisfying:

$$\mathbb{E}\left[\omega_t | W_{t-1}\right] = \mathbb{E}\left[\nabla \tilde{W}_l^t - \nabla W_l^t | W_{t-1}\right] = 0$$

**Lemma 2.** *Under the assmuptions in Theorem 1:*

$$\mathbb{E}\left[||\omega_t||^2 | W_{t-1}\right] < D$$

*for some constant $D$.*

*Proof.* We prove by induction. Since $\ell$ is $\beta$-Lipschitz, the gradients $\nabla y$ are bounded. During backpropagation the gradients are propagated by:

$$\nabla \tilde{a}_{l-1} = \texttt{approx}(W_l \Sigma'(h_l) \nabla \tilde{a}_l)$$

Let us assume $\nabla \tilde{a}_l$ is bounded and show that $\nabla \tilde{a}_{l-1}$ is bounded in expectation as well:

$$
\begin{aligned}
\mathbb{E}\left[||\nabla \tilde{a}_{l-1}||^2\right] &\leq \mathbb{E}\left[||\nabla \tilde{a}_{l-1} - W_l \Sigma'(h_l) \nabla \tilde{a}_l||^2\right] + \mathbb{E}\left[||W_l \Sigma'(h_l) \nabla \tilde{a}_l||^2\right] \\
&\leq C\, ||W_l||^2\, ||\Sigma'(h_l) \nabla \tilde{a}_l||^2 \\
&< D'
\end{aligned}
$$

for some constant $D'$, where the second inequality follows from the properties of `approx` and last inequality follows from the $\beta$-Lipschitz of $\Sigma$, the induction assumption on the boundness of $\nabla \tilde{a}_l$ and the assumption on the boundness of $W_l$.

The gradients $\nabla \tilde{W}$ are calculated by:

$$\nabla \tilde{W}_{l-1} = \texttt{approx}(a_{l-1} \Sigma'(h_l) \nabla \tilde{a}_l^\top)$$

and therefore:

$$
\begin{aligned}
\mathbb{E}\left[||\omega_t||^2 | W_{t-1}\right] &= \mathbb{E}\left[\left|\left|\nabla \tilde{W}_l^t - \nabla W_l^t\right|\right|^2 | W_{t-1}\right] \\
&= \mathbb{E}\left[\left|\left|\nabla \tilde{W}_l^t - (a_{l-1}\Sigma'(h_l)\nabla \tilde{a}_l^\top) + (a_{l-1}\Sigma'(h_l)\nabla \tilde{a}_l^\top) + \nabla W_l^t\right|\right|^2 | W_{t-1}\right] \\
&\leq \mathbb{E}\left[\left|\left|\nabla \tilde{W}_l^t - a_{l-1}\Sigma'(h_l)\nabla \tilde{a}_l^\top\right|\right|^2\right] + \mathbb{E}\left[||a_{l-1}\Sigma'(h_l)\nabla \tilde{a}_l^\top||^2\right] + \mathbb{E}\left[||\nabla W_l^t||^2\right] \\
&\leq C_1\, ||a_{l-1}||^2\, \mathbb{E}\left[||\Sigma'(h_l)\nabla \tilde{a}_l^\top||^2\right] + C_2\, ||a_{l-1}||^2\, \mathbb{E}\left[||\Sigma'(h_l)\nabla \tilde{a}_l^\top||^2\right] + \mathbb{E}\left[||\nabla W_l^t||^2\right] \\
&\leq D
\end{aligned}
$$

In the second inequality we used the properties of `approx`. In the last inequality we used the boundness of $\Sigma'$, $\nabla W_l^t$ from the assumptions, the boundness of $\mathbb{E}\left[||\nabla \tilde{a}_l^\top||^2\right]$ from above. In addition, we assumed boundness of the activations $a_{l-1}$. This assumption holds if the activation function $\sigma$ is bounded (for example, sigmoid) and in the general case it also requires the assumptions on the boundness of weights and inputs. $\square$

The same arguments can be made for the bias and the approximate bias gradients.

Based on Lemmas 1 and 2 and using standard analysis of SGD (for example [56] and [50]) the SGD convergence guarantees hold for approximate backpropagation as well. $\square$

**Remark.** *Both CRS and Bernoulli-CRS satisfy the property*

$$\mathbb{E}\left[\left|\left|A^\top B - \texttt{approx}(A^\top B)\right|\right|^2\right] \leq C\,||A||^2\,||B||^2$$

*since the expected Frobenius norm for the error matrix satisfies:*

$$\mathbb{E}\left[\left|\left|A^\top B - \tilde{A}^\top\tilde{B}\right|\right|_F^2\right] =$$

$$\frac{1}{k}\left(\sum_{i=1}^n e_i|A_{(i)}||B_{(i)}|\right)^2 - \sum_{i=1}^n|A_{(i)}|^2|B_{(i)}|^2$$

$$\leq \left(\sum_{i=1}^n e_i|A_{(i)}||B_{(i)}|\right)^2$$

$$\leq \left(\sum_{i=1}^n|A_{(i)}|^2\right)\left(\sum_{i=1}^n|B_{(i)}|^2\right)$$

$$= ||A||_F^2\,||B||_F^2$$

*where we used Theorem 3 and the Cauchy-Schwarz inequality.*

**Corollary.** *Let $f(x, W, b)$ be a multi-layer neural network with bounded $\beta$-Lipschitz activation functions $\sigma$. Let $\ell$ be a $\beta$-Lipschitz loss function, and let the network be trained with SGD using properly decreasing learning rate. If the weight gradient matrix products in the backward pass are approximated using an unbiased approximation scheme satisfying:*

$$\mathbb{E}\left[A^\top B - \texttt{approx}(A^\top B)\right] = 0$$

*and:*

$$\mathbb{E}\left[\left|\left|A^\top B - \texttt{approx}(A^\top B)\right|\right|^2\right] \leq C\,||A||^2\,||B||^2$$

*for some finite constant $C$ and some norm $||\cdot||$,*

*then then the approximated gradients are unbiased with bounded second moments.*

*Proof.* Lemma 1 under these assumptions holds by the same arguments. We now prove the equivalent of Lemma 2:

$$\mathbb{E}\left[||\omega_t||_F^2\,|W_{t-1}\right] = \mathbb{E}\left[\left|\left|\nabla\tilde{W}_l^t - \nabla W_l^t\right|\right|_F^2\,|W_{t-1}\right]$$

$$= \mathbb{E}\left[\left|\left|\nabla\tilde{W}_l^t - a_{l-1}\Sigma'(h_l)\nabla a_l^\top\right|\right|_F^2\right]$$

$$\leq ||a_{l-1}||_F^2\,\left|\left|\Sigma'(h_l)\nabla a_l^t\right|\right|_F^2$$

$$\leq D$$

The first inequality follows from the properties of `approx`. The second inequality follows from the $\beta$-Lipschitz property of $\ell, \Sigma$ bounding the second term, and from the boundness of the activation function $\sigma$ bounding the first term. $\qquad\square$

### C.2 Proofs for Section 6 - Sampling Without Scaling and Top-$k$ Selection

**Theorem 2.** *Let $A$ be a $n \times m$ random matrix and $B$ be $n \times p$ random matrix, such that*

$$\mathbb{E}\left[A^\top(i)B_{(i)}\right] = 0$$

*for $1 \leq i \leq n$. Assume $k$ column-row pairs with indices $\{j\}_1^n$ are sampled from $A$ and $B$.*

*Then, the MMSE estimator for the matrix product $A^\top B$ would be $\tilde{A}^\top\tilde{B}$ where $\tilde{A}, \tilde{B}$ are constructed from the sampled column-row pairs without scaling.*

*Furthermore, if $A^{\top(i)}B_{(i)}$ and $A^{\top(j)}, B_{(j)}$ are independent for different $i$ and $j$ then the MSE will be minimized when sampling $k$ pairs with the maximum norm multiplication $|A_{(i)}||B_{(i)}|$.*

*Proof.* Given sampled pairs $j_1, ..., j_k$ the MMSE estimator would be:

$$
\begin{aligned}
\widehat{A^\top B} &= \mathbb{E}\left[A^\top B | A_{(j_1)}, ..., A_{(j_k)}, B_{(j_1)}, ..., B_{(j_k)}\right] \\
&= \mathbb{E}\left[\sum_{i=1}^k A_{(j_i)}^\top B_{(j_i)} + \sum_{i \notin \{j\}_1^k} A^{\top(i)} B_{(i)} | A_{(j_1)}, ..., A_{(j_k)}, B_{(j_1)}, ..., B_{(j_k)}\right] \\
&= \sum_{i=1}^k A_{(j_i)}^\top B_{(j_i)} + \sum_{i \notin \{j\}_1^k} \mathbb{E}\left[A^{\top(i)} B_{(i)}\right] \\
&= \sum_{i=1}^k A_{(j_i)}^\top B_{(j_i)} \\
&= \tilde{A}^\top \tilde{B}
\end{aligned}
$$

The MSE would be:

$$
\mathbb{E}\left[\left\|A^\top B - \tilde{A}^\top \tilde{B}\right\|_F^2\right] = \mathbb{E}\left[\left\|\sum_{i \notin \{j\}_1^k} A^{\top(i)} B_{(i)}\right\|_F^2\right]
$$

if we assume independence between different column-row pairs $A^{\top(i)} B_{(i)}, A^{\top(j)} B_{(j)}$ then the last expression reduces to:

$$
\sum_{i \notin \{j\}_1^k} \mathbb{E}\left[\left\|A^{\top(i)} B_{(i)}\right\|_F^2\right] = \sum_{i \notin \{j\}_1^k} \mathbb{E}\left[|A_{(i)}|^2 |B_{(i)}|^2\right]
$$

and therefore will be minimized for a top-$k$ selection scheme that samples the pairs with the highest norm. $\qquad\square$

### C.3 Proofs for Section A - Bernoulli-CRS

**Proposition 1.** $\mathbb{E}\left[\tilde{A}^\top \tilde{B}\right] = A^\top B$

*Proof.*

$$
\begin{aligned}
\mathbb{E}\left[A^\top PKKPB\right] &= A^\top PP\mathbb{E}\left[KK\right] B \\
&= A^\top PP\mathbb{E}\left[K\right] B \\
&= A^\top B
\end{aligned}
$$

where we used that fact that $K$ is diagonal and that $K_{i,i} \in \{0, 1\}$. $\qquad\square$

**Proposition 2.** $\mathbb{E}\left[T\right] = k$

*Proof.*

$$
\mathbb{E}\left[T\right] = \mathbb{E}\left[\sum_{j=1}^n K_{j,j}\right] = \sum_{j=1}^n \mathbb{E}\left[K_{j,j}\right] = \sum_{j=1}^n p_j = k
$$

$\qquad\square$

**Proposition 3.**

$$
\mathbf{Var}\left[(\tilde{A}^\top \tilde{B})_{i,j}\right] = \sum_{t=1}^n \frac{1 - p_t}{p_t} A_{t,i}^2 B_{t,j}^2
$$

*Proof.* Fix $i, j$. From Proposition 1:

$$\mathbb{E}\left[(\tilde{A}^\top \tilde{B})_{i,j}\right] = (A^\top B)_{i,j}$$

Calculating the second moment:

$$
\begin{aligned}
\mathbb{E}\left[(\tilde{A}^\top \tilde{B})_{i,j}^2\right] &= \mathbb{E}\left[\left(\sum_{t=1}^n A_{t,i}\frac{K_{t,t}}{p_t}B_{t,j}\right)^2\right] \\
&= \mathbb{E}\left[\sum_{t=1}^n \sum_{u=1}^n A_{t,i}\frac{K_{t,t}}{p_t}B_{t,j}A_{u,i}\frac{K_{u,u}}{p_u}B_{u,j}\right] \\
&= \mathbb{E}\left[\sum_{t=1}^n \sum_{u\neq t}^n A_{t,i}\frac{K_{t,t}}{p_t}B_{t,j}A_{u,i}\frac{K_{u,u}}{p_u}B_{u,j}\right] \\
&\quad + \mathbb{E}\left[\sum_{t=1}^n A_{t,i}^2 B_{t,j}^2 \frac{K_{t,t}}{p_t^2}\right] \\
&= \sum_{t=1}^k \sum_{u\neq t}^k A_{t,i}B_{t,j}A_{u,i}B_{u,j} + \sum_{t=1}^n \frac{1}{p_t}A_{t,i}^2 B_{t,j}^2 \\
&= (A^\top B)_{i,j}^2 - \sum_{t=1}^n A_{t,i}^2 B_{t,j}^2 + \sum_{t=1}^n \frac{1}{p_t}A_{t,i}^2 B_{t,j}^2 \\
&= (A^\top B)_{i,j}^2 + \sum_{t=1}^n \frac{1-p_t}{p_t}A_{t,i}^2 B_{t,j}^2
\end{aligned}
$$

Therefore:

$$
\begin{aligned}
\mathbf{Var}\left[(\tilde{A}^\top \tilde{B})_{i,j}\right] &= \mathbb{E}\left[(\tilde{A}^\top \tilde{B})_{i,j}^2\right] - \mathbb{E}\left[(\tilde{A}^\top \tilde{B})_{i,j}\right]^2 \\
&= \sum_{t=1}^k \frac{1-p_t}{p_t}A_{t,i}^2 B_{t,j}^2
\end{aligned}
$$

$\square$

**Theorem 3.** *The expected Frobenius norm of the error matrix* $\mathbb{E}\left[\left\|\left|A^\top B - \tilde{A}^\top \tilde{B}\right\|\right|_F^2\right]$ *is* $\sum_{t=1}^n \frac{1-p_t}{p_t}|A_{(t)}|^2|B_{(t)}|^2$.

*Furthermore, under the constraint* $\sum_{i=1}^n p_i = k$ *it is minimized for the probabilities:*

$$p_i = \frac{|A_{(i)}||B_{(i)}|}{\sqrt{\mu}}\mathbb{1}_{\{0<|A_{(i)}||B_{(i)}|<\mu\}} + \mathbb{1}_{\{|A_{(i)}||B_{(i)}|\geq\mu\}}$$

*where* $\mu$ *is a root of the following function:*

$$G(\mu) := \sum_{i=1}^n \left(\frac{|A_{(i)}||B_{(i)}|}{\sqrt{\mu}}\mathbb{1}_{\{0<|A_{(i)}||B_{(i)}|<\mu\}} + \mathbb{1}_{\{|A_{(i)}||B_{(i)}|\geq\mu\}}\right) - k$$

*Proof.* Note:

$$
\begin{aligned}
\mathbb{E}\left[\left\|\left|A^\top B - \tilde{A}^\top \tilde{B}\right\|\right|_F^2\right] &= \sum_{i=1}^m \sum_{j=1}^p \mathbb{E}\left[\left(A^\top B - \tilde{A}^\top \tilde{B}\right)_{i,j}^2\right] \\
&= \sum_{i=1}^m \sum_{j=1}^p \mathbf{Var}\left[\left(\tilde{A}^\top \tilde{B}\right)_{i,j}\right]
\end{aligned}
$$

Therefore, using Proposition 3 we get:

$$\mathbb{E}\left[\left\|\left|A^\top B - \tilde{A}^\top \tilde{B}\right\|\right|_F^2\right] = \sum_{i=1}^{m}\sum_{j=1}^{p}\sum_{t=1}^{n} \frac{1-p_t}{p_t} A_{t,i}^2 B_{t,j}^2$$

$$= \sum_{t=1}^{n} \frac{1-p_t}{p_t}\left(\sum_{i=1}^{m} A_{t,i}^2\right)\left(\sum_{j=1}^{p} B_{t,j}^2\right)$$

$$= \sum_{t=1}^{n} \frac{1-p_t}{p_t}|A_{(t)}|^2|B_{(t)}|^2$$

Let us now find the optimal sampling probabilities that minimize the Frobenius error. Define the function:

$$f(p_1, p_2, ..., p_n) = \sum_{t=1}^{n} \frac{1-p_t}{p_t}|A_{(t)}|^2|B_{(t)}|^2$$

We can now consider the optimization problem:

$$\min_{p_1,...,p_n} \quad f(p_1, ..., p_n)$$
$$\text{s.t} \quad p_i - 1 \le 0$$
$$- p_i \le 0$$
$$\sum_{i=1}^{n} p_i - k = 0$$

We define the Lagrangian as:

$$L(p_1, ..., p_n, \lambda_1, ..., \lambda_n, \nu_1, ..., \nu_n, \mu) \triangleq$$
$$f(p_1, p_2, ..., p_k) + \sum_{i=1}^{n} \lambda_i (p_i - 1) - \sum_{i=1}^{n} \nu_i p_i + \mu\left(\sum_{i=1}^{n} p_i - k\right)$$

where $\lambda_i \ge 0$, $\nu_i \ge 0$ and $\mu \in \mathbb{R}$.

Applying KKT stationarity condition:

$$0 = \frac{\partial}{\partial p_i} L = -\frac{1}{p_i^2}|A_{(i)}|^2|B_i|^2 + \lambda_i - \nu_i + \mu = 0$$

Therefore:

$$p_i = \frac{|A_{(i)}||B_{(i)}|}{\sqrt{\lambda_i - \nu_i + \mu}}$$

Next we divide into 3 cases,
**Case 1: If** $p_i \in (0, 1)$**:** In this case due to complementary-slackness we obtain $\lambda_i = \nu_i = 0$, and therefore,

$$p_i = \frac{|A_{(i)}||B_{(i)}|}{\sqrt{\mu}}$$

**Case 2: If** $p_i = 1$**:** In this case due to complementary-slackness we obtain $\nu_i = 0$, and therefore,

$$1 = p_i = \frac{|A_{(i)}||B_{(i)}|}{\sqrt{\mu + \lambda_i}}$$

**Case 3: If** $p_i = 0$**:** In this case due to complementary-slackness we obtain $\lambda_i = 0$, which implies that,

$$0 = p_i = \frac{|A_{(i)}||B_{(i)}|}{\sqrt{\mu - \nu_i}}$$

but this can only happen if $|A_{(i)}||B_{(i)}| = 0$.

Combining the above we conclude that given $\mu$ one can write the solution as follows,

$$p_i = \frac{|A_{(i)}||B_{(i)}|}{\sqrt{\mu}}1_{\{0<|A_{(i)}||B_{(i)}|<\mu\}} + 1_{\{|A_{(i)}||B_{(i)}|\geq\mu\}}$$

Now, in order to satisfy the equality conditions $\mu$ should satisfy the following equality,

$$\sum_{i=1}^{n}\left(\frac{|A_{(i)}||B_{(i)}|}{\sqrt{\mu}}1_{\{0<|A_{(i)}||B_{(i)}|<\mu\}} + 1_{\{|A_{(i)}||B_{(i)}|\geq\mu\}}\right) = k$$

Now, one can actually find $\mu$ using bisection, To see this consider the following function,

$$G(\mu) := \sum_{i=1}^{n}\left(\frac{|A_{(i)}||B_{(i)}|}{\sqrt{\mu}}1_{\{0<|A_{(i)}||B_{(i)}|<\mu\}} + 1_{\{|A_{(i)}||B_{(i)}|\geq\mu\}}\right) - k$$

And note that $G(\mu)$ is a one dimensional monotonically decreasing (actually non-increasing) function of $\mu$.

Also, if we sorts the $|A_{(i)}||B_{(i)}|$'s, i.e. $|A_{(1)}||B_{(1)}| \leq |A_{(2)}||B_{(2)}| \leq \ldots |A_{(n)}||B_{(n)}|$, then given $j$ such that $\mu \in (|A_{(j)}||B_{(j)}|, |A_{(j+1)}||B_{(j+1)}|)$, then we can find the exact value of $\mu$ from the equality constraints equation:

$$\sum_{i=1}^{n}\left(\frac{|A_{(i)}||B_{(i)}|}{\sqrt{\mu}}1_{\{0<|A_{(i)}||B_{(i)}|<\mu\}} + 1_{\{|A_{(i)}||B_{(i)}|\geq\mu\}}\right) = k$$

$\square$

**Corollary.** *The sampling probabilities*

$$p_i = \min\left\{\frac{k|A_{(i)}||B_{(i)}|}{\sum_{j=1}^{n}|A_{(j)}||B_{(j)}|}, 1\right\}$$

*are optimal if* $k \leq \frac{\sum_{i=1}^{n}|A_{(i)}||B_{(i)}|}{\max_i|A_{(i)}||B_{(i)}|}$

*Proof.* As a simpler, sub-optimal solution for the above optimization problem we propose the following relaxation. First, we solve the optimization problem without the inequality conditions:

$$0 \leq p_i \leq 1$$

Then, for each optimal $p_i^*$ we clamp the value between the range $[0, 1]$. This allows us to comply with the inequality conditions that allows to treat $p_i$ as a parameter to Bernoulli distribution at the expense of relaxing the constraint on the sum of the parameters $p_i$, leading to potentially sub-optimal solution.

As the first step, we therefore solve the problem:

$$\min_{p_1,\ldots,p_n} \quad f(p_1, \ldots, p_n)$$

$$\text{s.t} \quad \sum_{i=1}^{n} p_i - k$$

To minimize $f$ subject to the constraint $\sum_{i=1}^{n} p_i = k$ we use the Lagrange multiplier $\lambda$ and define the function:

$$g(p_1, p_2, \ldots, p_n) = f(p_1, p_2, \ldots, p_n) + \lambda\left(\sum_{i=1}^{n} p_i - k\right)$$

Deriving and equaling to zero we get:

$$0 = \frac{\partial g}{\partial p_i} = -\frac{1}{p_i^2}|A_{(i)}|^2|B_i|^2 + \lambda$$

Therefore:
$$p_i = \frac{|A_{(i)}||B_{(i)}|}{\sqrt{\lambda}}$$

Substituting in $\sum_{i=1}^{n} p_i = k$:
$$\sum_{i=1}^{n} \frac{|A_{(i)}||B_{i)}|}{\sqrt{\lambda}} = k$$
$$\sqrt{\lambda} = \frac{\sum_{i=1}^{n}|A_{(i)}||B_{(i)}|}{k}$$

And therefore we get:
$$p_i = \frac{k|A_{(i)}||B_{(i)}|}{\sum_{i=1}^{n}|A_{(i)}||B_{(i)}|}$$

And the final result after clamping would be:
$$p_i = \min\left\{ \frac{k|A_{(i)}||B_{(i)}|}{\sum_{i=1}^{n}|A_{(i)}||B_{(i)}|}, 1 \right\}$$

Note that this solution yields $p_i \geq 0$, satisfying one of the original inequality conditions. What about $p_i \leq 1$?

If
$$k \leq \frac{\sum_{i=1}^{n}|A_{(i)}||B_{(i)}|}{\max_i |A_{(i)}||B_{(i)}|}$$

then the second inequality conditions holds as well and the solution is indeed the optimal solution to the original problem.

Substituting in the expression for the Frobenius error we get:

$$\mathbb{E}\left[\left\|A^\top B - \tilde{A}^\top \tilde{B}\right\|_F^2\right] = \sum_{t=1}^{n} \frac{1 - p_t}{p_t}|A_{(t)}|^2|B_{(t)}|^2$$
$$= \frac{1}{k}\left(\sum_{i=1}^{n}|A_{(i)}||B_{(i)}|\right)^2 - \sum_{i=1}^{n}|A_{(i)}|^2|B_{(i)}|^2$$

$\square$

The following theorem yields high probability bounds for the Frobenius and spectral norms for the Bernoulli-CRS algorithm:

**Theorem 4.** *Let $A \in \mathbb{R}^{n \times m}$ and $B \in \mathbb{R}^{n \times p}$. Let $\tilde{A}, \tilde{B}$ be the sampled matrices according to the Bernoulli-CRS algorithm described above. Denote*
$$R \triangleq \max_i \left\|A^{\top (i)} B_{(i)}\right\|$$

*and*
$$\sigma^2 \triangleq \frac{1}{k}\left(\sum_{i=1}^{n} e_i |A_{(i)}||B_{(i)}|\right)^2 - \sum_{i=1}^{n} e_i |A_{(i)}|^2|B_{(i)}|^2$$

*then, for all $t \geq 0$:*

$$\mathbb{P}\left\{\left\|A^\top B - \tilde{A}^\top \tilde{B}\right\| \geq t\right\} \leq (m + p) \cdot \exp\left(\frac{-t^2/2}{\sigma^2 + Rt/3}\right)$$

$$\mathbb{P}\left\{\left\|A^\top B - \tilde{A}^\top \tilde{B}\right\|_F \geq t\right\} \leq (m + p)^{3/2} \cdot \exp\left(\frac{-t^2/2}{\sigma^2 + Rt/3}\right)$$

*Proof.* The Matrix Bernstein concentration inequality states:

**Theorem** (**Matrix Bernstein [57]**). *Consider a finite sequence $\{\mathbf{Z}_k\}$ of independent, random matrices with dimensions $d_1 \times d_2$. Assume that each random matrix satisfies*

$$\mathbb{E}\left[\mathbf{Z}_k\right] = 0 \quad and \quad ||\mathbf{Z}_k|| \leq R \quad almost\ surely.$$

*Define*

$$\sigma^2 \triangleq \max\left\{\left\|\sum_k \mathbb{E}\left[\mathbf{Z}_k\mathbf{Z}_k^\top\right]\right\|, \left\|\sum_k \mathbb{E}\left[\mathbf{Z}_k^\top\mathbf{Z}_k\right]\right\|\right\}$$

*Then, for all $t \geq 0$,*

$$\mathbb{P}\left\{\left\|\sum_k \mathbf{Z}_k\right\| \geq t\right\} \leq (d_1 + d_2) \cdot \exp\left(\frac{-t^2/2}{\sigma^2 + Rt/3}\right)$$

.

In our sampling algorithm, we can define:

$$\mathbf{Z}_k \triangleq A_{(k)}^\top B_{(k)} - \frac{1}{p_k} K_{k,k} A_{(k)}^\top B_{(k)}$$

when $K_{k,k}$ is a Bernoulli random variable with parameter $p_k$ as defined above. It is clear that $\mathbb{E}\left[\mathbf{Z}_k\right] = 0$.

Also, let us define:

$$R \triangleq \max_k \left\|A_{(k)}^\top B_{(k)}\right\|$$

so it it also clear that $||\mathbf{Z}_k|| \leq R$.

By construction, $\{\mathbf{Z}_k\}$ are independent.

We can also define:

$$\sigma^2 \triangleq \max\left\{\left\|\sum_k \mathbb{E}\left[\mathbf{Z}_k\mathbf{Z}_k^\top\right]\right\|, \left\|\sum_k \mathbb{E}\left[\mathbf{Z}_k^\top\mathbf{Z}_k\right]\right\|\right\}$$

$$= \max\left\{\left\|\mathbb{E}\left[\sum_k \mathbf{Z}_k\mathbf{Z}_k^\top\right]\right\|, \left\|\mathbb{E}\left[\sum_k \mathbf{Z}_k^\top\mathbf{Z}_k\right]\right\|\right\}$$

$$= \max\left\{\left\|\mathbb{E}\left[(A^\top B - \tilde{A}^\top\tilde{B})(A^\top B - \tilde{A}^\top\tilde{B})^\top\right]\right\|,\right.$$

$$\left.\left\|\mathbb{E}\left[(A^\top B - \tilde{A}^\top\tilde{B})^\top(A^\top B - \tilde{A}^\top\tilde{B})\right]\right\|\right\}$$

$$\leq \max\left\{\mathrm{Tr}\left(\mathbb{E}\left[(A^\top B - \tilde{A}^\top\tilde{B})(A^\top B - \tilde{A}^\top\tilde{B})^\top\right]\right),\right.$$

$$\left.\mathrm{Tr}\left(\mathbb{E}\left[(A^\top B - \tilde{A}^\top\tilde{B})^\top(A^\top B - \tilde{A}^\top\tilde{B})\right]\right)\right\}$$

$$= \mathbb{E}\left[\left\|A^\top B - \tilde{A}^\top\tilde{B}\right\|_F^2\right]$$

$$= \frac{1}{k}\left(\sum_{i=1}^n e_i |A_{(i)}||B_{(i)}|\right)^2 - \sum_{i=1}^n e_i |A_{(i)}|^2|B_{(i)}|^2$$

where we used the linearity of expectation and trace, the property $||A|| \leq \mathrm{Tr}[A]$ for positive semidefinite matrices and the expected Frobenius norm from Theorem 3.

The bound on the spectral norm follows immediately from the Matrix Berenstein inequality.

Using the property:

$$||A||_F \leq \sqrt{r}\,||A||$$

we get the similar result for the Frobenius norm, factored by $\sqrt{m+p}$. $\qquad\square$

## C.4 Proofs for Section B - Approximating Convolutions

The following proofs go along the same lines of [20], generalizing them to multi-channel convolutions (zero-padding assumed).

**Lemma 3.** *Suppose* $I \in \mathbb{R}^{IW \times IC}_{B \times IH}, K \in \mathbb{R}^{IC \times OC}_{KW \times KW}, 1 \le k \le IC, \{p_i\}^{IC}_{i=1}$ *is a probability distribution over* $\{1, ..., IC\}$ *and* $\{i_t\}^k_{t=1}$ *are such that* $i_t \in \{1, ..., IC\}$.

*Let* $O \in \mathbb{R}^{OW \times OC}_{B \times OH} = I * K$ *be the multi-channel convolution of* $I, K$ *as defined in* (28) *and let* $\tilde{O}$ *be its approximation by sampling* $k$ *input channels as defined in* (30). *Then:*

$$\mathbb{E}\left[\tilde{O}\right] = O$$

*Proof.* We show that every $b, oh, ow, oc$ satisfies $\mathbb{E}\left[\tilde{O}^{ow,oc}_{b,oh}\right] = O^{ow,oc}_{b,oh}$.

For $t \in \{1, ..., k\}$, define $X_t = (\frac{I^{[i_t]} * K_{[i_t]}}{p_{i_t}})^{ow,oc}_{b,oh}$.

Using (30) we can write $\tilde{O}_{b,oh,ow,oc} = \sum^k_{t=1} \frac{1}{k} X_t$.

Taking the expectation, we get:

$$(\mathbb{E}\left[\tilde{O}\right])^{ow,oc}_{b,oh} = \mathbb{E}\left[\sum^k_{t=1} \frac{1}{k} X_t\right] = \mathbb{E}\left[X_t\right] = \sum^{IC}_{i=1} p_i \cdot \frac{(I^{[i]} * K_{[i]})^{ow,oc}_{b,oh}}{p_i} = O^{ow,oc}_{b,oh} \qquad (32)$$

$\square$

**Lemma 4.** *Suppose the same as Lemma 3. Then:*

$$\mathbf{Var}\left[\tilde{O}^{ow,oc}_{b,oh}\right] = \frac{1}{k} \sum^{IC}_{i=1} \frac{1}{p_i} \sum^{KH}_{h=1} \sum^{KW}_{w=1} (I^{ow+w-1,i}_{b,oh+h-1})^2 (K^{i,oc}_{h,w})^2$$
$$+ \frac{1}{k} \sum^{IC}_{i=1} \frac{1}{p_i} \sum_{\substack{h,h'=1 \\ h \ne h'}}^{KH} \sum_{\substack{w,w'=1 \\ w \ne w'}}^{KW} I^{ow+w-1,i}_{b,oh+h-1} I^{ow+w'-1,i}_{b,oh+h'-1} K^{i,oc}_{h,w} K^{i,oc}_{h',w'}$$
$$- \frac{1}{k} (O^{ow,oc}_{b,oh})^2$$

*Proof.* Define $X_t$ as in Lemma 3. From (30) and the independence of different $X_t$:

$$\mathbf{Var}\left[\tilde{O}^{ow,oc}_{b,oh}\right] = \mathbf{Var}\left[\sum^k_{t=1} \frac{1}{k} X_t\right] = \frac{1}{k} \mathbf{Var}\left[X_t\right] = \frac{1}{k}(\mathbb{E}\left[X^2_t\right] - \mathbb{E}\left[X_t\right]^2) \qquad (33)$$

$$\mathbb{E}\left[X^2_t\right] = \sum^{IC}_{i=1} p_i \cdot \frac{((I^{[i]} * K_{[i]})^{ow,oc}_{b,oh})^2}{p^2_i}$$
$$= \sum^{IC}_{i=1} \frac{1}{p_i} \left(\sum^{KH}_{h=1} \sum^{KW}_{w=1} I^{ow+w-1,i}_{b,oh+h-1} K^{i,oc}_{h,w}\right)^2 \qquad (34)$$

From (32) we get $\mathbb{E}\left[X_t\right] = O$.

Substituting both expressions in (33) and expanding concludes the proof. $\square$

**Lemma 5.** *Suppose the same as Lemma 3. Then:*

$$\mathbb{E}\left[\|O - \tilde{O}\|^2_F\right] = \sum^{IC}_{i=1} \frac{\|I^{[i]}\|^2_F \cdot \|K_{[i]}\|^2_F - E^i_{IK} + R^i_{IK}}{kp_i} - \frac{1}{k}\|O\|^2_F$$

*where*

$$E_{IK}^i = \sum_{b=1}^{B} \sum_{\substack{oh,ow\ s.t \\ oh<KH\ or \\ ow<KW}} \sum_{oc=1}^{OC} \sum_{h=1}^{KH} \sum_{\substack{h,w\ s.t \\ h>oh\ or \\ w>ow}}^{KH,KW} (I_{b,oh}^{ow,i})^2 (K_{h,w}^{i,oc})^2$$

$$R_{IK}^i = \sum_{b=1}^{B} \sum_{oh=1}^{OH} \sum_{ow=1}^{OW} \sum_{oc=1}^{OC} \sum_{\substack{h,h'=1 \\ h\neq h'}}^{KH} \sum_{\substack{w,w'=1 \\ w\neq w'}}^{KW} I_{b,oh+h-1}^{ow+w-1,i} I_{b,oh+h'-1}^{ow+w'-1,i} K_{h,w}^{i,oc} K_{h',w'}^{i,oc}$$

*The expected error is minimized when the sampling probabilities are:*

$$p_i = \frac{\sqrt{\left\|I^{[i]}\right\|_F^2 \cdot \left\|K_{[i]}\right\|_F^2 - E_{IK}^i + R_{IK}^i}}{\sum_{j=1}^{IC} \sqrt{\left\|I^{[j]}\right\|_F^2 \cdot \left\|K_{[j]}\right\|_F^2 - E_{IK}^j + R_{IK}^j}}$$

**Remark.** *We use here the Frobenius norm in its generalization for tensors. For a tensor T of rank r:*

$$\|T\|_F = \sqrt{\sum_{j_1,j_2,\ldots,j_r} T_{j_1,j_2,\ldots,j_r}^2}$$

*Proof.* Note that:

$$\mathbb{E}\left[\left\|O - \tilde{O}\right\|_F^2\right] = \sum_{b=1}^{B} \sum_{oh=1}^{OH} \sum_{ow=1}^{OW} \sum_{oc=1}^{OC} \mathbb{E}\left[((O - \tilde{O})_{b,oh}^{ow,oc})^2\right] = \sum_{b=1}^{B} \sum_{oh=1}^{OH} \sum_{ow=1}^{OW} \sum_{oc=1}^{OC} \mathbf{Var}\left[\tilde{O}_{b,oh}^{ow,oc}\right]$$

Substituting the result from Lemma 4:

$$\mathbb{E}\left[\left\|O - \tilde{O}\right\|_F^2\right] = \sum_{i=1}^{IC} \frac{1}{kp_i} \sum_{b=1}^{B} \sum_{oh=1}^{OH} \sum_{ow=1}^{OW} \sum_{oc=1}^{OC} \sum_{h=1}^{KH} \sum_{w=1}^{KW} (I_{b,oh+h-1}^{ow+w-1,i})^2 (K_{h,w}^{i,oc})^2$$

$$+ \sum_{i=1}^{IC} \frac{1}{kp_i} \sum_{b=1}^{B} \sum_{oh=1}^{OH} \sum_{ow=1}^{OW} \sum_{oc=1}^{OC} \sum_{\substack{h,h'=1 \\ h\neq h'}}^{KH} \sum_{\substack{w,w'=1 \\ w\neq w'}}^{KW} I_{b,oh+h-1}^{ow+w-1,i} I_{b,oh+h'-1}^{ow+w'-1,i} K_{h,w}^{i,oc} K_{h',w'}^{i,oc} \qquad (35)$$

$$- \frac{1}{k} \sum_{b=1}^{B} \sum_{oh=1}^{OH} \sum_{ow=1}^{OW} \sum_{oc=1}^{OC} (O_{b,oh}^{ow,oc})^2$$

This expression includes 3 terms. The first involves products between each element of $I^{[i]}$ and all the corresponding entries in $K_{[i]}$, except for the upper and left edges of $I^{[i]}$. We therefore add and subtract the correction term $E_{IK}^i$ to get:

$$\sum_{i=1}^{IC} \frac{1}{kp_i} \sum_{b=1}^{B} \sum_{oh=1}^{OH} \sum_{ow=1}^{OW} \sum_{oc=1}^{OC} \sum_{h=1}^{KH} \sum_{w=1}^{KW} (I_{b,oh+h-1}^{ow+w-1,i})^2 (K_{h,w}^{i,oc})^2$$

$$= \sum_{i=1}^{IC} \frac{1}{kp_i} \left( \left( \sum_{b=1}^{B} \sum_{oh=1}^{OH} \sum_{ow=1}^{OW} (I_{b,oh+h-1}^{ow+w-1,i})^2 \right) \left( \sum_{oc=1}^{OC} \sum_{h=1}^{KH} \sum_{w=1}^{KW} (K_{h,w}^{i,oc})^2 \right) - E_{IK}^i \right)$$

$$= \sum_{i=1}^{IC} \frac{\left\|I^{[i]}\right\|_F^2 \cdot \left\|K_{[i]}\right\|_F^2 - E_{IK}^i}{kp_i}$$

The second term is $\sum_{i=1}^{IC} \frac{1}{kp_i} R_{IK}^i$.

The third term can be written as $\frac{1}{k} \sum_{b=1}^{B} \sum_{oh=1}^{OH} \sum_{ow=1}^{OW} \sum_{oc=1}^{OC} (O_{b,oh,ow,oc})^2 = \frac{1}{k} \|O\|_F^2$

Substituting these terms in (35) yields the result of (5).

To find $\{p_i\}_{i=1}^{IC}$ that minimize the expression in (5) it is enough to minimize the function $f = \sum_{i=1}^{IC} \frac{\alpha_i^2}{p_i}$ under the constraints $\sum p_i = 1$ and $p_i > 0$. We can write the numerator as $\alpha_i^2$ because the expression in (34) is non-negative.

This minimization problem has a straightforward solution in Lemma 4 of [20], which is $p_i = \frac{\alpha_i}{\sum_{j=1}^{IC} \alpha_j}$.

In our case, $\alpha_i = \sqrt{\left\|I^{[i]}\right\|_F^2 \cdot \left\|K_{[i]}\right\|_F^2 - E_{IK}^i + R_{IK}^i}$, and therefore the optimal probabilities are:

$$p_i = \frac{\sqrt{\left\|I^{[i]}\right\|_F^2 \cdot \left\|K_{[i]}\right\|_F^2 - E_{IK}^i + R_{IK}^i}}{\sum_{j=1}^{IC} \sqrt{\left\|I^{[j]}\right\|_F^2 \cdot \left\|K_{[j]}\right\|_F^2 - E_{IK}^j + R_{IK}^j}}$$

The terms $E_{IK}^i, R_{IK}^i$ emerge for convolutions when the kernel spatial dimensions are greater than one. However, computing them is too expensive, precluding efficient implementation of the approximate version. We therefore omit them and verify empirically whether the resulting norm-proportional probabilities:

$$p_i = \frac{\left\|I^{[i]}\right\|_F \cdot \left\|K_{[i]}\right\|_F}{\sum_{j=1}^{IC} \left\|I^{[j]}\right\|_F \cdot \left\|K_{[j]}\right\|_F}$$

yield better results than the uniform sampling. Intuitively, in some (common) cases these terms are much smaller than $\left\|I^{[i]}\right\|_F^2 \cdot \left\|K_{[i]}\right\|_F^2$, so their omission does not significantly impact the final outcome. $E_{IK}^i$ amounts to the outer spatial dimensions of the input not being convolved with the entire kernel, so it is likely to be smaller than the Frobenius norm of the whole input. $R_{IK}^i$ is the sum of products of different input and kernel entries. If different kernels are lowly-correlated and weights are centered around zero, the sum will include terms of similar magnitudes but opposite signs.

$\square$

## D    Implementation Details

All single-node results were obtained using 2.2GHz Intel Xeon Silver 4210 CPU with four NVidia V100 GPUs with 32GB of memory. Wall-time speedup were measured when running with a single GPU, except ResNet-152 where 2 GPUs are used due to memory capacity. We used PyTorch version 1.7.0 with CUDA 10.1 and Python version 3.6.9.

### D.1    MLP for MNIST

The MNIST dataset [28] includes 60K training examples and 10K test examples. We use 5K as validation set. Each example is a $28 \times 28$ gray-scale image of a handwritten digit.

Our MLP model contains the following layers:

- $784 \times 500$ fully-connected layer with RELU activations.
- $500 \times 500$ fully-connected layer with RELU activations.
- $500 \times 10$ fully-connected layer with RELU activations.
- Log Softmax

We use the Adam optimizer [58] with default parameters (learning rate=0.001,$\beta_1 = 0.9$,$\beta_2 = 0.999$,$\epsilon = 1e-08$). As loss function we use negative log likelihood. We use minibatch size of 50 and train the model for 20 epochs.

We apply sampling to all the fully connected layers. When sampling in the backward pass, we do not reduce the batch dimension below 10 in the weight gradient computation.

Figure 7(a) shows the MNIST test accuracy for different sampling algorithms and sampling ratios in the forward pass. We observe that top-$k$ performs the best. Figure 7(b) shows the same when approximations are applied in the backward pass only. In this case, all sampling algorithms are similar when performing above 30% of the backward pass computations.

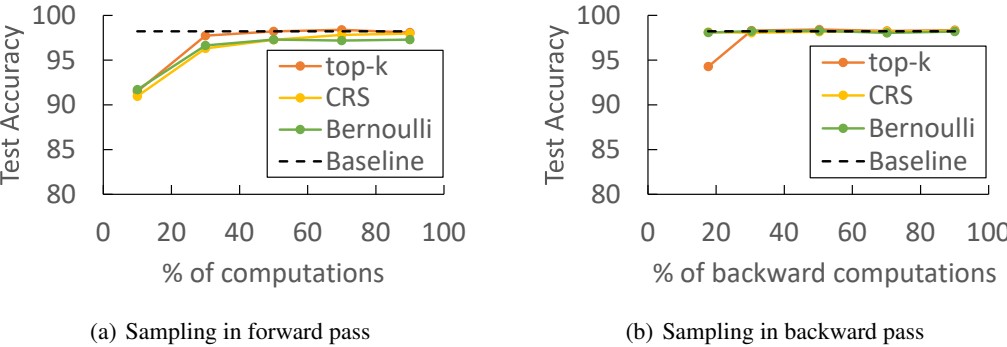

(a) Sampling in forward pass

(b) Sampling in backward pass

Figure 7: MNIST test accuracy for MLP, under different approximating algorithms and different sampling ratios

## D.2    CNN for MNIST

The network is composed of the following layers:

- $5 \times 5 \times 32$ convolution layer with RELU activation, followed by $2 \times 2$ max pooling.
- $5 \times 5 \times 64$ convolution layer with RELU activation, followed by $2 \times 2$ max pooling.
- Dropout layer with $p = 0.5$.
- $3136 \times 1024$ fully connected layer with RELU activation.
- $1024 \times 10$ fully connected layer.
- Dropout layer with $p = 0.5$.
- Log Softmax

The model is trained using Adam optimizer with default parameters (learning rate=0.001,$\beta_1 = 0.9$,$\beta_2 = 0.999$,$\epsilon = 1e - 08$) and negative log likelihood loss. We use minibatch size of 50 and train the model for 20 epochs.

We apply sampling to the convolutional layers. When sampling in the backward pass, we do not reduce the batch dimension below 10 in the weight gradient computation.

Figure 8(a) shows the MNIST test accuracy for different sampling algorithms and sampling ratios in the forward pass. We observe that top-$k$ performs the best. Figure 8(b) shows the same when approximations are applied in the backward pass only. In this case, all sampling algorithms are similar when performing above 30% of the backward pass computations.

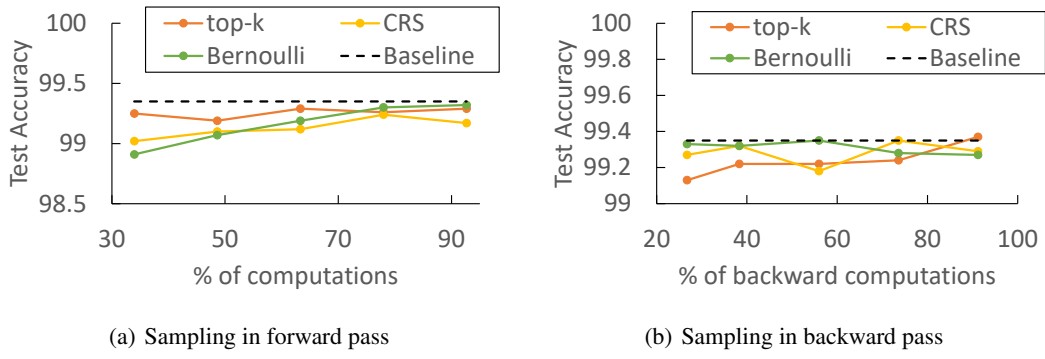

(a) Sampling in forward pass

(b) Sampling in backward pass

Figure 8: MNIST test accuracy for CNN, under different approximating algorithms and different sampling ratios

### D.3 Wide ResNet-28-10 for CIFAR-10

The CIFAR-10 dataset [30] consists of $32 \times 32$ color images from 10 classes, split into 50K training set and 10K test set.

For WRN-28-10 [29] we use the implementation in `https://github.com/meliketoy/wide-resnet.pytorch`, aviable under MIT License.

WRN-28-10 includes the following layers:

- conv1 - $3 \times 3 \times 16$ input convolution layer
- conv2 - eight $3 \times 3 \times 160$ convolution layers
- conv3 - eight $3 \times 3 \times 320$ convolution layers
- conv4 - eight $3 \times 3 \times 640$ convolution layers
- Batch normalization, $8 \times 8$ Average pooling, fully connected+softmax layers.

Every two subsequent convolution layers are followed by a residual connection that adds the input to these layers to the result. the first convolution conv3 and conv4 has a stride of 2, halving the spatial dimensions. For additional details see [29].

Image preprocessing includes padding to 36x36 and random crop, horizontal flipping and per-image whitening. The optimizer is Momentum SGD with momentum=0.9 and 5e-4 weight decay. Learning rate is 0.15 for the first 60 epochs, 0.03 until epoch 120, 0.006 until epoch 160 and 0.0012 afterwards. We use batch size of 256, cross-entropy loss and train the model for 200 epochs.

We apply sampling to the convolutional layers except the first layer due to the small number of input channels (3) and the single fully-connected layer which amounts only to 0.01% of the total computations in WRN-28-10. When sampling in the backward pass, we do not reduce the batch dimension below 10 in the weight gradient computation.

Figure 9(a) shows the CIFAR-10 test accuracy for different sampling algorithms and sampling ratios in the forward pass. We observe that top-$k$ performs the best. Figure 9(b) shows the same when approximations are applied in the backward pass only. In this case, Bernoulli-CRS performs the best but is still below 1% of the baseline accuracy until 90% sampling ratio.

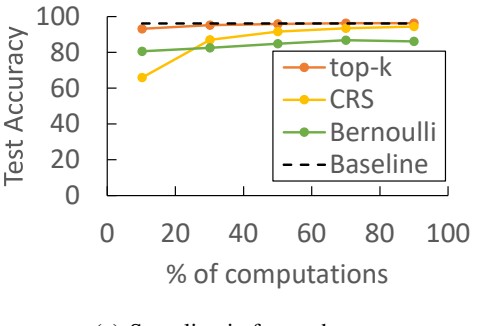

(a) Sampling in forward pass

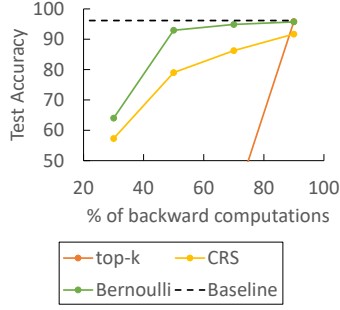

(b) Sampling in backward pass

Figure 9: CIFAR-10 test accuracy for WRN-28-10, under different approximating algorithms and different sampling ratios

Figure 6(a) shows the CIFAR-10 validation accuracy learning curves for different forward-pass top-$k$ sampling ratios, compared to the non-approximate baseline. We observe that higher sampling ratios lead to slower learning at the early training stages but the gap is decreasing as the training progresses. Figure 10 focuses on the last training epochs to observe the accuracies in more detail. We observe that 50% sampling is slightly lower than the non-approximate baseline, while less aggressive approximations that perform 70% or 90% of the computations achieve identical or slightly higher validation accuracy.

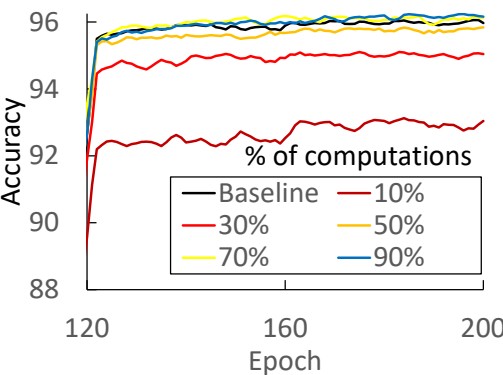

Figure 10: Learning curves for WRN-28-10 CIFAR-10 validation accuracy under different top-$k$ sampling ratios. Focused view of last training epochs

### D.4 ResNet-50 and ResNet-152 for ImageNet

The ImageNet [32] ILSVRC 2012 dataset contain 1.2 million training images of varying dimensions split into 1000 classes. The validation set includes 50K images and the test set consists of 100K images.

For ResNet-50 [31] we use the implementation in `https://github.com/pytorch/examples/tree/master/imagenet`, available under BSD 3-Clause License. See [31] for further details on ResNet-50 architecture.

Image preprocessing includes random 224x224 crop, horizontal flipping and image normalization. The optimizer is Momentum SGD with momentum=0.9 and 1e-4 weight decay. Learning rate is 0.1 and it is decayed by 10 every 30 epochs. We use batch size of 256, cross-entropy loss and train the model for 90 epochs.

We apply sampling to the convolutional layers except the first layer due to the small number of input channels (3) and the fully-connected layer.

Figure 11(a) shows the top-1 accuracy of ResNet-50 for different sampling ratios. The different data points correspond to 50% top-$k$ sampling applied to all the layers, all layers with at least 128 channels, 256 channels, 512 channels and 1024 channels.

Figure 11(b) shows the top-1 accuracy of ResNet-152 for different sampling ratios. The different data points correspond to 50% top-$k$ sampling applied to all the layers, all layers with at least 256 channels, 512 channels and 1024 channels.

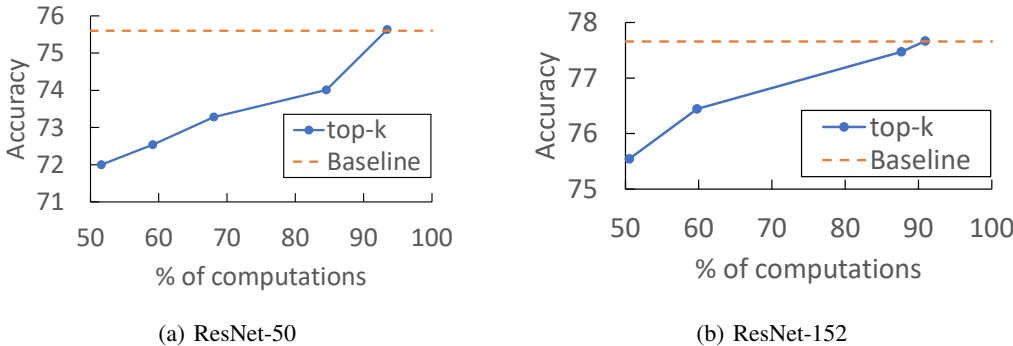

(a) ResNet-50

(b) ResNet-152

Figure 11: ResNet-50 and ResNet-152 ImageNet top-1 test accuracy. The accuracy increases as higher amounts of the computations are performed

Figures 6(b) and 6(c) show the top-1 validation accuracy learning curves for different forward-pass top-$k$ sampling ratios, compared to the non-approximate baseline. We observe that ResNet-50 and ResNet-152 are more sensitive to sampling compared to WRN-28-10 on CIFAR10. Nonetheless, applying 50% sampling in the layers with 1024 channels, corresponding to 93% of the computations in ResNet-50 and 91% of the computations in ResNet-152, follow the non-approximate learning curves almost identically.

### D.5 Distributed Training

To evaluate the accuracy of top-$k$-weights algorithm for ResNet-152 on Imagenet we used the same settings as in the previous section and trained on a single node with 4 GPUs. The accuracy results are shown in figure 12. The different data points correspond to 50% top-$k$-weights sampling applied to all layers with at least 256 channels, 512 channels and 1024 channels.

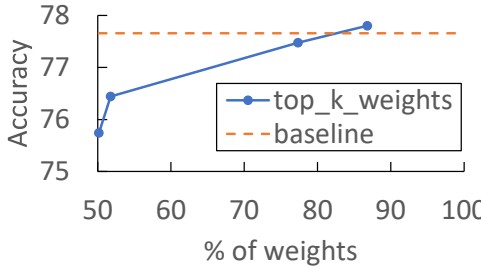

Figure 12: ResNet-152 ImageNet top-1 test accuracy, using top-$k$-weights algorithm.

For the distributed training experiments we used eight AWS EC2 instances equipped with 2.7GHz Intel Xeon E5-2686v4 CPU, one V100 GPU with 16 GB of memory, 10 Gbps networking, PyTorch version 1.7.1, CUDA 11 and Python 3.7.6.

For the distributed measurement we used the same hyper-parameters except the minibatch size which we set to 32 per GPU. We could not increase the batch size since the AWS EC2 GPU we used had 16GB of memory and could not support higher batch size. We note that the eight-node setting has a total global batch size of 256, which matches the batch size used in the accuracy evaluation.