# OpenReview forum: "Faster Neural Network Training with Approximate Tensor Operations"
_NeurIPS.cc/2021/Conference — NeurIPS 2021 Poster_

### Official Review · Reviewer_pWRo · 2021-07-14

**Rating:** 6
**Confidence:** 4

**Summary:**

This paper proposes a method for speeding up training by approximating the computation of linear and convolutional layers. The approximation of such operation is done by taking K samples of the operands (e.g. K row-column pairs in a matrix multiplication) instead of multiplying them directly. The authors motivate the use of the CRS algorithm because is lightweight, can be applied to matrices/tensors of all shapes and, it is unbiased. The authors then study its use in Linear Regression settings and Convolutional layers and, propose their own Bernoulli-based sampling strategy. The authors evaluate their approximate training scheme on common image classification tasks, including a distributed learning scenario.

**Limitations And Societal Impact:**

- Limited evaluation of the proposed method.
- No clear advantage over existing techniques (e.g. gradient sparsification) that are evaluated on standard image classification tasks.

**Main Review:**

This paper touches upon an important problem that arises in two radically different settings: first when training massive models, even when hyperparamters (e.g. batch-size, number of nodes, etc) are correct, being able to approximate the compute could give that extra 1.5x  to make the experiment finish in a more reasonable time frame; second, when training small or medium size models on very constrained devices like in wearables, which are both compute and memory restricted and (!) battery powered. In essence both scenarios are similar in the sense that both push the hardware to the limit. As authors point out, there are other ways of introducing “approximations”, being quantization the most widely used in practice.

The paper is very well written, the notation is easy to follow. As the authors refer in their work, there have been a number of works proposing ways of speeding up training by using approximations. Two works that weren’t cited but are relevant are [1], [2].

To revise/improve/expand:

- My main concerns are regarding the length of the evaluation section and number of experiments in general.  Given the number of other works attempting to also speedup training, I would say that the proposed method, which emphasises on *sampling*, would benefit from evaluating scenarios/models/datasets where *sampling* would have an advantage over the other techniques.
- Someone could argue that the model architectures chosen (WRN-28-10 for CIFAR-10 or ResNet-152) are overparameterized for each tasks, making it easier for the algorithm proposed to show higher compute reduction rations. It would be interesting to show how the technique proposed in this work could speedup networks that have already been pruned or quantized.
- (minor) I would advice improving the captions of the figures and tables to make them more self-contained. For example, what are the differences between each of the rows for ResNet-152 @ 8 nodes?
- Could the method proposed in this paper be interpreted as a (unstructured) pruning technique? Specially when using Top-K selection. Could the authors comment upon this?
- (for the future), the space of efficient training is getting busy. Looking into specific (instead of generic image classification workload) areas where ML is currently limited could be a good way to making the sampling technique here propose really shine. Going back to the previous paragraph, I think this technique could be valuable for on-device training in scenarios with limited compute/memory capabilities (e.g. federated learning on wearables/phones)

[1] [ReSprop](https://openaccess.thecvf.com/content_CVPR_2020/html/Goli_ReSprop_Reuse_Sparsified_Backpropagation_CVPR_2020_paper.html)

[2] [Backprop with Approximate Activations](https://arxiv.org/abs/1901.07988)

### Update
Post rebuttal update, I raise my score from 4 to 6. (see comment below for reasons).

**Time Spent Reviewing:**

4

---

> ### Author Response · Authors · 2021-08-10
> **Author Response to Reviewer pWRo**
>
> Thank you for the review. The feedback and criticism are valuable and valid. We would still like to reply on some of the issues raised, and hope that the virtues of our work will outweigh its limitations.
>
> $\bf{Comment}$: Missing references to relevant works: “ReSprop” and “Backprop with Approximate Activations”
>
> $\bf{Response}$: Thank you for the references, we will add them.
>
> $\bf{Comment}$: Short experimental section and insufficient experimental work.
>
> $\bf{Response}$: Please see the “Implementation Details” section in the supplementary material, containing more thorough experimental work around different sampling algorithms, sampling ratios, forward vs backward sampling and training plots. We did not include that in the paper body due to space limitations.
>
> $\bf{Comment}$: It is possible that the speedups obtained are due to overparametrization.
>
> $\bf{Response}$:  Overparametrization is common in neural networks and was shown also to have theoretical advantages (for example [1]). Even if our method works well because of overparametrization, it would still be useful in cases the network is indeed overparametrized, or in scenarios like architecture search before the network is pruned and quantized for efficient inference. To be honest, our choice of wide networks was mostly due to the better wall-time speedups obtained. Narrower layers benefit less from sampling due to GPU overheads.
>
> $\bf{Comment}$: No clear benefits over other methods such as quantization and pruning.
>
> $\bf{Response}$: We do not propose sampling as a replacement for other acceleration methods. We try to show that it is a useful addition to the DL acceleration toolbox, and other acceleration techniques can be complementary. Indeed, quantization and pruning can accelerate the training. Our approach is orthogonal to these and in principle can be combined.
>
> $\bf{Comment}$: What are the differences between each of the rows for ResNet-152 @ 8 nodes?
>
> $\bf{Response}$: The difference is the percentage of the overall multiply-accumulate operations and gradient communication traffic that are avoided due to sampling. In the first row the sampling is more aggressive, resulting in more savings but lower accuracy. In the last row there is no accuracy loss at all, but the savings and speedup are more modest. The supplementary material (D5) provides more details on the sampling ratio and on which layers it was applied.
>
> $\bf{Comment}$: Could this method be interpreted as a pruning technique?
>
> $\bf{Response}$: Since we sample in each iteration depending on both weights and activations, this is not necessarily pruning in its classical form. inputs with different meaningful features/channels will lead to sampling different weights but the model size and expression capacity will remain higher than a pruned model. This is by the way a potential advantage of our method over classical quantization/pruning techniques that impose constraints on the entire model and might limit its expressiveness. In other words, our method can be interpreted as "minibatch-adaptive pruning" where the global model size remains the same but different subsets of the network are used to best match the current minibatch.
>
> That said, the authors of [2] (which extends the [3] work we cite) noticed that in some cases sampling gradients leads to consistently learning the same subset of weights, and in that case the model can be pruned to remove the weights that are seldom updated. We did not evaluate this direction in this work and focused on accelerating training. We believe that the potential pruning benefits for inference need to be compared to other pruning methods targeting inference.
>
> $\bf{Comment}$: It is worth exploring the benefits of this methods in specialized areas where it could shine such as federated learning in resource-constrained environments.
>
> $\bf{Response}$: Good idea, thank you. Having a poster-child application will make a stronger case. We still hope that by showing good results on general workloads we can make the case that our method is a valid and useful contribution to the DL training acceleration field.
>
> [1] Du, S. and Lee, J., 2018, July. On the power of over-parametrization in neural networks with quadratic activation. In International Conference on Machine Learning (pp. 1329-1338). PMLR.
>
> [2] Sun et al, 2018. Training simplification and model simplification for deep learning: A minimal effort back propagation method. IEEE Transactions on Knowledge and Data Engineering, 32(2), pp.374-387.
>
> [3] Sun et al., 2017, July. meprop: Sparsified back propagation for accelerated deep learning with reduced overfitting. In International Conference on Machine Learning (pp. 3299-3308). PMLR.

---

> > ### Comment · Reviewer_pWRo · 2021-08-23
> > **Raising score 4->6**
> >
> > I would like to thank the authors for their effort in addressing the concerns and clarifications I mentioned in my review. I would advice attempting to include more of the results in the main body of the paper instead of offloading most of them to the Appendix. I'm raising my score from 4 to 6 because I think approximate computations is an important new are for optimising ML, which come with its own challenges. The community has spent many years trying to approximate operands (e.g. quantization, sparsity, etc) but not so much making the core algorithm (e.g. matmul, convolution) more lightweight since they are normally computed in an exact way. I think approximate operations is one way to enable that.

---

> > > ### Author Response · Authors · 2021-08-27
> > > **Thanks. We will attempt including more experimental results in the paper body.**
> > >
> > > We thank reviewer pWRo for the comments and updated score.
> > >
> > > Due to the 8 page limit we had to compromise between the paper body and the supplementary material. We would like to use the additional camera-ready content page to move experimantal results from the appendix to the main body.

---

### Official Review · Reviewer_yvhy · 2021-07-15

**Rating:** 6
**Confidence:** 4

**Summary:**

The paper proposes to approximate matrix multiplications (and convolutions) by randomly sampling column-row pairs and only multiplying submatrices. This is used to compute stochastic gradients which subsequently speed up training. Different sampling modes are presented and compared in experiments. Theoretical and practical training speed-ups are reported on standard architectures and datasets.

**Limitations And Societal Impact:**

Sufficiently addressed (limitations: see above)

**Main Review:**

The paper is very well written and easy to follow. It considers an important problem and the idea of speeding up training in the proposed way is (to my knowledge) a novel and worthy contribution.

Reducing training time is especially important on datasets on which training takes very long. Therefore, more results on ImageNet are required. Results from the appendix using top-k show that accuracy degrades substantially for slight theoretical reductions in training time (e.g., Fig. 10a). There seem to be no results for CRS/Bernoulli on ImageNet. Figure 1a suggests that it might be possible that CRS/Bernoulli provides a better approximation for lower computational budgets. This needs to be verified. Furthermore, it would be interesting to see if more epochs would result in the original accuracy, e.g., twice as many epochs at 50% computations per iteration.

How can you explain that Eq. (12) produces worse results although it appears to be unbiased? The only way I see is that it substantially increases variance. It would be interesting to verify this and put a short comment on that in text. The same is true for the method presented in Section 5 where only the backward pass is approximated, which does not seem to produce better results (see lines 281/282).

Line 192 says that the estimates are unbiased. However, line 197 says that there is an additional regularization term induced. In my opinion you cannot really consider this to be an unbiased estimate of the original loss function then.

Line 137 says that improved accuracies due to regularization were not observed. However, it is later shown that the induced loss function obtained for Bernoulli sampling has an additional regularization term (Eq. (17)). This is somehow contradictory and requires clarification. In fact, the induced regularizer (17) reminds me of results from [1]. What is the difference to [1].

Line 165 says that the Expectation E[S_jj S_tt] is not equal to 1 because S_jj and S_tt are not independent. I do not immediately see why independence would imply that the expectation equals 1. Could you please clarify that for me.

I am missing some kind of summary / takeaway message of the findings in the conclusion section?

Minor:
- line 150: single dot "." looks strange

[1] Wager et al.; Dropout Training as Adaptive Regularization; NIPS 2013

**Time Spent Reviewing:**

5

---

> ### Author Response · Authors · 2021-08-10
> **Author Response to Reviewer yvhy**
>
> Thank you very much for the review, time and effort. We will try below to address the comments raised.
>
> $\bf{Comment}$: No results with CRS/Bernoulli on ImageNet
>
> $\bf{Response}$: Indeed, we did not experiment with CRS/Bernoulli on ImageNet because of their worse performance on CIFAR10 compared to top-k.
>
> $\bf{Comment}$:  Figure 1a suggests that CRS/Bernoulli might perform better on smaller sampling ratio
>
> $\bf{Response}$: We did experiment with CRS/Bernoulli with lower sampling ratio on CIFAR10, as reported in the supplementary material (figure 8a). Our experiments indicate that CRS/Bernoulli are consistently worse than top-k which better matches Figure 1b.
>
> $\bf{Comment}$: Will more epochs result in the original accuracy?
>
> $\bf{Response}$: We did some early experiments along this direction and did not observe that. However, we did not combine that with a thorough hyperparameter tuning so we cannot currently provide a definitive answer. We believe future research might find such a tradeoff.
>
> $\bf{Comment}$: Why does Eq. (12) provide worse results despite being unbiased? If the reason is increased variance, it is worth verifying and commenting on that.
>
> $\bf{Response}$: Good point. We agree that increased variance is the plausible explanation, and we will add that.
>
> $\bf{Comment}$: It is inaccurate to refer to Bernoulli Sampling as unbiased in line 192 because line 197 says it adds a regularization.
>
> $\bf{Response}$: Line 192 is unrelated to linear regression. It refers to the fact that stand-alone matrix multiplication using Bernoulli Sampling provides an unbiased estimate of the original product. Line 197 refers to linear regression and in that case training with Bernoulli Sampling under SGD provides unbiased gradient estimate of the  regularized objective, which is a biased estimate of the original objective. We will change the wording in line 196 to better clarify that.
>
> $\bf{Comment}$: Line 137 says that improved accuracies due to regularization were not observed. It contradicts Eq. (17) which shows a regularization term.
>
> $\bf{Response}$: Eq. (17) shows in theory that in the simple case of linear regression sampling is equivalent to regularization. Line 137 mentions that in practice, we did not observe accuracy improvements from that. While slightly disappointing for us, it is not a strict contradiction – regularization does not always improve accuracy, and there might be cases we did not observe where our methods will improve accuracy and not just training speed.
>
> $\bf{Comment}$: What is the difference between the derivation of regularization for Bernoulli-CRS with a similar derivation for Dropout in [1]?
>
> $\bf{Response}$: Thank you for this reference, we were unaware of it and will add it to the related work. Indeed, the results are similar, and the main difference is that in [1] the Dropout/sampling probability is assumed to be a constant p for all elements, while our results are more general by allowing a different sampling probability per element, resulting in the different probabilities showing up in the regularization term itself.
>
> $\bf{Comment}$: Line 165: Why would independence of S_jj and S_tt imply that the expectation equals 1?
>
> $\bf{Response}$: Please see Eq.(3) which in other words states that the expectation of $\tilde{S}$ is the identity matrix, i.e the expectation of the diagonal elements is 1. Therefore, if different diagonal elements are independent their multiplication will be 1 as well since $E[XY]=E[X]E[Y]$ for independent $X,Y$.
>
> $\bf{Comment}$: Missing summary and takeaway in the conclusions section.
>
> $\bf{Response}$: Agreed, we did not add that due to space limit and would like to elaborate more in the camera ready. The main takeaway as we see it is that we believe that sampling based approximations and fast approximations in general are valuable additions for the DL acceleration “toolbox” and a fertile ground for further research.
>
> $\bf{Comment}$: Single dot in line 150 looks strange.
>
> $\bf{Response}$: Right, we will fix that.
>
> [1] Wager et al.; Dropout Training as Adaptive Regularization; NIPS 2013

---

> > ### Comment · Reviewer_yvhy · 2021-08-30
> > **Response**
> >
> > I would like to thank the authors for addressing my concerns and clarifying questions. I remain with my original score.

---

### Official Review · Reviewer_nbsG · 2021-07-16

**Rating:** 6
**Confidence:** 3

**Summary:**

This paper adapts the CRS method for approximate matrix multiplication into a new method, Bernoulli-CRS, that yields unbiased gradient estimates. The paper then analyzes this method on linear regression and shows the new method and giving unbiased estimates of a gradient for the original objective with a regularization term. They then devise a new routine for approximate backpropagation that has unbiased estimates with bounded second moments. The paper then introduces a deterministic variant called top-k sampling and show the performance of the 3 methods (CRS, Bernoulli-CRS, top-k) in experiments in terms of performance, compute, and speed.

**Limitations And Societal Impact:**

Limitations - There's a discrepancy between practice and theory where it seems practically you need to approximate both directions but the theory only supports the backward pass.

**Main Review:**

Originality - A somewhat novel adaptation to a pre-existing method that addresses some of the problem to applying it in a deep learning context.

Quality - The theory for approximate backpropagation seem applicable to practice and justify the use of this method. However, its not immediately clear whether it applies to ResNets, which is the only network architecture used on larger datasets. Furthermore, there is no justification for the use of this method in the forward pass, so it would be good to have some explanation for why it approximating the forward and backward pass performs so well, and in particular, why it outperforms only approximating the backward pass.

Clarity - Overall the presentation is clear and organized. However, the section describing the practical applicability of Theorem 2 is unclear. While the theorem is easily applicable at initialization with independent zero-mean entries, its not clear what it means after any number of iterations. In particular, the backpropagation of a gradient signal should introduce dependencies between the weights, so the pairwise-independence is almost immediately violated.
As well, in line 230 the phrase "if the distribution remains centered around zero during training" isn't entirely clear to me. Given that standard NN training only takes a point initialization and not a distribution around each weight, I'm not sure what it means to be "centered around zero during training". The discussion in the experiments section seems to imply something like the distribution generated by the trajectory of the weight, but it's not clear.
The conclusions reached in the linear regression sections aren't totally clear. The CRS method was deemed suboptimal since it would not yield unbiased gradient estimates; however, the Bernoulli-CRS was concluded to have the property of an expected gradient equal to one for linear regression with an additional regularization term -- so this too is not an unbiased gradient estimate. It seems you may also be able to recast the expected gradient from CRS as a linear regression + regularization gradient (but I didn't do the math so may it doesn't exactly work out). Because of that, it's not clear why one method is preferable to the other in this linear regression toy problem.

Significance - The experimental results show promising speedups with little loss in performance. This can be particularly useful in cases where there are many networks being trained to find the best, such NAS or an ablation study. Some more analysis about the effects of Bernoulli-CRS would be required before standard use of the method, as there may be implicit regularization effects by this method (as seen in the linear regression case) and maybe other effects that aren't obvious. The main reason for worry is that the only approximate the backward pass caused big performance degradation in the larger networks even though that seems quite well understood by the theory; on the other hand, approximate the forward as well improved performance but is not really understood theoretically.

Minor Note - In equations 5, 11, 15, should the sum go up to n rather than k?

**Time Spent Reviewing:**

5

---

> ### Author Response · Authors · 2021-08-10
> **Author Response for Reviewer nbsG**
>
> Thank you for your review. Below we try to address some of the comments and questions raised.
>
> $\bf{Comment}$: It is unclear whether the theory for approximate backpropagation applies to ResNets
>
> $\bf{Response}$: While we formulate Theorem 1 for simple feed-forward networks, the proof in the supplementary material only relies on the linearity of back propagation and the chain rule. Since backpropagation through ResNets is linear as well, we believe the extension of the result to ResNets is relatively simple.
>
> $\bf{Comment}$: There's a discrepancy between practice and theory. No theoretical justification to approximation in the forward pass and why it outperforms approximate backpropagation in practice.
>
> $\bf{Response}$: Acknowledged. There is a gap between what worked for us in practice and what we were able to theoretically prove. We invested significant efforts in trying to narrow this gap but were unable to prove general results for forward sampling in deep non-linear networks. Sections 3 and 4 try to provide intuition by showing that in linear regression forward sampling is equivalent to regularization, but it is still far from a rigorous proof for the non-linear case. This gap opens a window for further theoretical investigations, and we hope that future works will succeed in bridging it.
>
> $\bf{Comment}$: Theorem 2 is not applicable in practice because weights and inputs are not pairwise independent during training
>
> $\bf{Response}$: It is true that weights and inputs are not independent in the strict sense, but if the correlation between them is low (“almost independent”) it could justify why top-k sampling would yield good results. The reference we provide for the pairwise-independent assumption, [1], shows empirically that this assumption does hold in some cases.
>
> $\bf{Comment}$: in line 230 the phrase "if the distribution remains centered around zero during training" is unclear because we do not assume a distribution around each weight.
>
> $\bf{Response}$: By “distribution” we do not mean a distribution around each weight, but rather the histogram/distribution obtained by looking at many different weights. The references for this claim in the paper show empirically that the weights histogram/distribution remains centered around zero during training, and we try to use that to empirically justify why top-k sampling works well in practice for neural network training.
>
> $\bf{Comment}$: It is unclear why Bernoulli-CRS is better than CRS for linear regression – both are biased, and if we interpret the bias as regularization it can also be applied to both methods.
>
> $\bf{Response}$: The main advantage of Bernoulli-CRS is that the bias can be interpreted as a variation of the well-known L2 regularization. We tried to do the math for CRS but the resulting bias terms were pretty nasty and we were not able to see how they can be mapped to a sensible regularization expression.
>
> $\bf{Comment}$: In equations (5), (11), (15) should the sum go up to n rather than k?
>
> $\bf{Response}$: Thank you for pointing this out. In equation (5) the sum should go up to k but the indices should be referenced like in equation (1). Equations (11) and (15) should indeed to up to n
>
> [1] Huang et al; Convolution-weight-distribution assumption: Rethinking the criteria of channel pruning

---

> > ### Comment · Reviewer_nbsG · 2021-08-30
> > **Reviewer Response**
> >
> > Thanks for the clarifications. I will maintain my score.

---

### Official Review · Reviewer_G6cy · 2021-07-16

**Rating:** 6
**Confidence:** 4

**Summary:**

This paper focuses on unbiased approximations to matrix multiplication and
convolution computations in deep neural networks (DNNs). The aim is to
reduce the complexity of these computations without the resulting function
the network as a whole computes. As practically all of the computation in
a deep network is expended in matrix multiplications and convolutions this
would provide a global acceleration. In practice this work demonstrates
4-37% improvement in training speed on ImageNet, depending on how the
computation is distributed.

**Limitations And Societal Impact:**

The experiments in this paper do not compare against other methods that
aim to reduce the complexity of matrix multiplications in deep learning.
The authors could compare against methods that replace the matrix
multiplications with other algorithms, such as [FFTs][] or [Tensor
networks][tt] or other methods listed in Section 2.1.

[ffts]: https://arxiv.org/abs/1511.05946
[tt]: https://epubs.siam.org/doi/abs/10.1137/090752286

**Main Review:**

## Originality

> Are the tasks or methods new? Is the work a novel
> combination of well-known techniques? (This can be valuable!) Is it clear
> how this work differs from previous contributions? Is related work
> adequately cited?

Approximating matrix multiplications with stochastic algorithms is a large
area of inquiry in numerical computing. The authors apply this to the
matrix multiplications that are performed during training of a deep neural
network to make training more efficient.

The authors provide a good review in Section 2.1 of related work in the are
of approximate matrix multiplication, not limited to stochastic algorithms.
It is clear that it would be useful to find out if the CRS algorithm can be
applied in training deep neural networks.

One of the main findings of this work is:

> ...we propose a new _Bernoulli-CRS_ variant which achieves statistical
> independence of samples, study its properties, and show that in linear
> regression it is equivalent to dynamic $L_2$ weight regularisation of the
> original, non-approximate loss.

However, the relationship between a Bernoulli noise mask (dropout) has been
known since [Fast Dropout][fast] in 2013. In addition, the method designed
in that work also was used to speed up training and appears to speed up
training more than 50% on MNIST, as presented here. However, they do not
present results on larger datasets as that was not expected at the time.

The extension to top-$k$ selection is novel to my knowledge. It is a
natural extension to the CRS approximation and is well explored in theory
and experiment by this work.

[fast]: http://proceedings.mlr.press/v28/wang13a.html

## Quality

> Is the submission technically sound? Are claims well supported
> (e.g., by theoretical analysis or experimental results)? Are the methods
> used appropriate? Is this a complete piece of work or work in progress? Are
> the authors careful and honest about evaluating both the strengths and
> weaknesses of their work?

The argument of the paper is delivered clearly. The use of CRS is well
motivated and argued in theory. The experiments then demonstrate that this
method can improve the efficiency of a neural network in training. It is
clear that the method should work and it does work.

The motivation behind the Top-$k$ selection algorithm to further improve
on CRS is sound. It is investigated in experiment and theory correctly. It
is clear that it works and is an improvement over CRS.

Section 7 describes how the algorithm may be extended to apply to
convolution. It seems unnecessary to include this as a common method to
implement convolution is the im2col-gemm algorithm and this involves a
matrix multiplication that CRS trivially applies to.

While the experiments show that the methods work, the authors do not
investigate the effect of hyperparameters. The training speed improvements
observed are relatively small and could be affected by learning rate,
regularisation, architecture. The stochastic algorithm itself could affect
convergence to the target accuracy by operating as a regulariser in a
similar way to dropout.

For example, in the [fast dropout][fast] paper the
authors specifically aimed to remove the random noise added by dropout
using better regularisation to speed up the experiment.

It is not clear to me that this training speedup would remain stable across
architectures or hyperparameter choices.

[fast]: http://proceedings.mlr.press/v28/wang13a.html

## Clarity

> Is the submission clearly written? Is it well organized? (If not, please
> make constructive suggestions for improving its clarity.) Does it
> adequately inform the reader? (Note that a superbly written paper provides
> enough information for an expert reader to reproduce its results.)

The ideas in this paper are presented in a natural sequence that is easy to
follow. The reader is able to follow easily the chain of reasoning that
leads from CRS to top-$k$ sampling for approximating matrices and why these
could be useful in deep learning.

Section 4 is titled "Bernoulli Sampling" and this could be misleading for
some readers because the algorithm being described is known as [Poisson
Sampling][poisson]. However, the name Poisson sampling may also be
misleading to readers that assume that samples will be drawn from a Poisson
distribution. Another name entirely may be more appropriate.

I feel confident I could reproduce the results presented in this paper
using the description in main body of the text.

[poisson]: https://en.wikipedia.org/wiki/Poisson_sampling

## Significance

> Are the results important? Are others (researchers or practitioners) likely
> to use the ideas or build on them? Does the submission address a difficult
> task in a better way than previous work?  Does it advance the state of the
> art in a demonstrable way? Does it provide unique data, unique conclusions
> about existing data, or a unique theoretical or experimental approach?

Universally improving the efficiency of matrix multiplications in deep
learning would be a valuable addition to the field. Unfortunately, it is
not clear to me that the methods presented in this paper are able to do
that.

However, an investigation of whether these methods can be used in deep
learning is valuable as a reference to future work. I think that this work
could provide unique results in that area that will be useful for future
researchers.

**Time Spent Reviewing:**

3

---

> ### Author Response · Authors · 2021-08-10
> **Author Response to Reviewer G6cy**
>
> Thank you for the detailed review. Below we try to address some of the comments and concerns raised.
>
> $\bf{Comment}$: No reference to “Fast Dropout” which is similar to the proposed “Bernoulli-CRS” and also accelerates MNIST training
>
> $\bf{Response}$: Thank you for the reference to Fast Dropout, we were not aware of it and will add it to the related work section. However, the Fast Dropout method seems substantially different from ours: they are not using sampling, and actually introduce overhead that makes each iteration 1.5x slower compared to regular dropout (section 6.3 in Fast Dropout paper), unlike ours. Their MNIST speedup comes from using L-BGFS optimization instead of SGD (section 6.3), and it is not clear that this method is applicable to the much larger networks we evaluate. We also derive theoretical results on “Bernoulli-CRS” which are different than Fast Dropout due to the different nature of the algorithms.
>
> $\bf{Comment}$: Convolution section is unnecessary because im2col translates convolutions to matrix multiplication
>
> $\bf{Response}$: Indeed, when im2col is used the results for approximate matrix multiplication trivially hold. However, our sampling algorithm for convolutions will also benefit other convolution algorithms that do not perform im2col (for example, gemm-based[1-2] or [3]) since the resulting tensors are smaller.
>
> $\bf{Comment}$: Speedup might be a result of hyperparameters and not sampling. Hyperparameters were not investigated, it is unclear if the speedup and accuracy would remain stable for different hyperparameters choices.
>
> $\bf{Response}$: Regarding sensitivity to hyperparameters, in our CIFAR-10 experiments we did try different hyperparameters and did not observe a significant change in behavior. Below is data from our log files (not included in the submission) which shows a similar behavior to what we included in table 1 for 50% top-k sampling for different learning rates, i.e training speedup with small accuracy loss:
>
> | Learning rate | topk test accuracy | baseline test accuracy |
>
> |     0.2       |       95.91       |        96.04\%          |
>
> |     0.18      |       95.71%       |        96.10%          |
>
> |     0.16      |       95.42%       |        96.05%          |
>
> |     0.14      |       95.78%       |        96.33%          |
>
> |     0.12      |       95.76%       |        96.13%          |
>
> |     0.1       |       95.75%       |        96.14%          |
>
> On the Imagenet studies we did not experiment with different hyperparameters due to the length of each run. For our approximate runs we simply used the same ‘default’ hyperparameters that were supplied for the non-approximate baselines. We see the fact that our algorithm maintains accuracy and shows speedup without additional hyperparameter tuning as a strength.
> The only difference between the approximate and non-approximate runs is the usage of sampling approximations, and therefore the wall-time speedup can only be attributed to that.
>
> $\bf{Comment}$: The name “Bernoulli Sampling” is misleading
>
> $\bf{Response}$: Acknowledged, thank you for pointing this out. We will change that to “Bernoulli-CRS” which is how we also refer to this algorithm in other places in the paper.
>
> $\bf{Comment}$: No comparison with other methods that aim to reduce the complexity of matrix multiplications in deep learning
>
> $\bf{Response}$: We are not aware of other methods that ours directly competes against (and is not complementary). Many acceleration methods are more relevant to inference while we target training. In section 2.1 we survey other potential methods but argue why they are not suitable for general usage in training which is our goal in this work.
>
> [1] Anderson et al; Low-memory gemm-based convolution algorithms for deep neural networks. arXiv preprint arXiv:1709.03395.
>
> [2] Georganas et al; Anatomy of high-performance deep learning convolutions on simd architectures. In SC18: International Conference for High Performance Computing, Networking, Storage and Analysis (pp. 830-841). IEEE.
>
> [3] Cho, M. and Brand, D., 2017, July. MEC: memory-efficient convolution for deep neural network. In International Conference on Machine Learning (pp. 815-824). PMLR.

---

> > ### Comment · Reviewer_G6cy · 2021-08-24
> > **Related Work Addressed, Competitiveness Open**
> >
> > I thank the authors for addressing my concern relating to prior work on the relationship between L2 regularisation and dropout. It is certainly difficult to compare the Fast Dropout experimental results to contemporary work given the differences in training practice.
> >
> > The authors also address my concern about related work on efficiency in deep learning. I do not know of another paper providing an in depth exploration of these approximate matrix multiplication aglorithms. It seems like an worthwhile addition to the literature for reference and I will update my score to reflect that.
> >
> > However, I don't think the experimental results showing stability over learning rate demonstrate that this approximation method is likely to provide efficiency benefits in general. The speed benefits are modest (up to 37%) which, while one can argue that it should be complementary to hyperparameter or architecture choice, in deep learning it is likely to affect convergence. Focused engineering of hyperparameter and architecture choice can speed up training and inference by massive factors, such as demonstrated in the [DAWNbench][] challenge, which has demonstrated improved single GPU training from 3 hours to 1 minute (approx 18,000% faster) on CIFAR-10. Example implementation can be found [here][davidcpage].
> >
> > One approximate matrix multiplication that this method is technically orthogonal to but may affect convergence would be mixed precision training. I would be surprised to see that training with top-k selection would not interfere with results using, for example, [Nvidia's supplied AMP][amp] which itself offers 200% improvements in training speed without affecting accuracy on ImageNet.
> >
> > [amp]: https://pytorch.org/blog/accelerating-training-on-nvidia-gpus-with-pytorch-automatic-mixed-precision/
> > [davidcpage]: https://github.com/davidcpage/cifar10-fast
> > [dawnbench]: https://dawn.cs.stanford.edu/benchmark/index.html#cifar10-train-time

---

> > > ### Author Response · Authors · 2021-08-27
> > > **Thanks, and trying to further address competitiveness**
> > >
> > > We thank reviewer G6cy for the response and updated score.
> > >
> > > We did perform some initial experiments which we believe indicate promising potential for combination with other approximation/acceleration techniques. For Resnet-152 with "top-k weights" algorithm using Nvidia's AMP on V100 and without further hyperparameter tuning we got the following results:
> > >
> > > | % gradients saved | accuracy loss |
> > >
> > > | 13% | 0.3% |
> > >
> > > | 23% | 0.7% |
> > >
> > > | 48% | 1.5% |
> > >
> > > The accuracy loss was relatively small, around 0.5% more compared to the FP32 training we reported in table 1. We believe that with hyperparameter tuning the gap would be even smaller. We note that for FP16 training the compute saving turned into lower wall-time speedups but we believe a dedicated kernel combining sampling and convolution/gemm could achieve high perfromance gains as well.

---

### Decision · Program_Chairs · 2021-09-27

**Decision:**

Accept (Poster)

**Comment:**

This paper presents a method for approximating the matrix multiplication and convolution operations in neural networks. Theoretical results show that the gradient estimates are robust to the approximation applied in the backwards pass (but not necessarily the forwards pass). Experiments show that the number of computations can be reduced significantly without degrading test accuracy.

Reviewers generally feel like the paper is sound overall, without significant gaps in correctness or discussion of prior work. They had various specific comments which they feel were mostly addressed in the author response. However, the reviewers kept their scores at middling values, due to concerns about whether the method translates into significant wall clock gains, whether it outperforms other approximations, and whether the noise robustness will extend to other architectures. These are all valid concerns, but this approach seems like worthwhile addition to the toolbox, and the authors have validated it in a reasonable variety of situations. I'd favor acceptance.